# Regularized Proportional Fairness Mechanism for Resource Allocation Without Money

**Sihan Zeng**                                                                                   *sihan.zeng@jpmchase.com*
*JPMorgan AI Research*

**Sujay Bhatt**                                                                                   *sujay.bhatt@jpmchase.com*
*JPMorgan AI Research*

**Alec Koppel**                                                                                   *alec.koppel@jpmchase.com*
*JPMorgan AI Research*

**Sumitra Ganesh**                                                                               *sumitra.ganesh@jpmorgan.com*
*JPMorgan AI Research*

**Reviewed on OpenReview:** *https://openreview.net/forum?id=K85nJ60lDR*

## Abstract

Mechanism design in resource allocation studies dividing limited resources among self-interested agents whose satisfaction with the allocation depends on privately held utilities. We consider the problem in a *payment-free* setting, with the aim of maximizing social welfare while enforcing incentive compatibility (IC), i.e., agents cannot inflate allocations by misreporting their utilities. The well-known proportional fairness (PF) mechanism achieves the maximum possible social welfare but incurs an undesirably high *exploitability* (the maximum unilateral inflation in utility from misreport and a measure of deviation from IC). In fact, it is known that no mechanism can achieve the maximum social welfare and exact incentive compatibility (IC) simultaneously without the use of monetary incentives (Cole et al., 2013). Motivated by this fact, we propose learning an *approximate* mechanism that desirably trades off the competing objectives. Our main contribution is to design an innovative neural network architecture tailored to the resource allocation problem, which we name Regularized Proportional Fairness Network (`RPF-Net`). `RPF-Net` regularizes the output of the PF mechanism by a learned function approximator of the most exploitable allocation, with the aim of reducing the incentive for any agent to misreport. We derive generalization bounds that guarantee the mechanism performance when trained under finite and out-of-distribution samples and experimentally demonstrate the merits of the proposed mechanism compared to the state-of-the-art.

The PF mechanism acts as an important benchmark for comparing the social welfare of any mechanism. However, there exists no established way of computing its exploitability. The challenge here is that we need to find the maximizer of an optimization problem for which the gradient is only implicitly defined. We for the first time provide a systematic method for finding such (sub)gradients, which enables the evaluation of the exploitability of the PF mechanism through iterative (sub)gradient ascent.

## 1 Introduction

Mechanism design studies how to *allocate* items (resources) to market participants (*agents*) holding private preferences with the goal of maximizing a criterion such as cumulative revenue (Manelli & Vincent, 2007; Navabi & Nayyar, 2018; Kazumura et al., 2020) or social welfare (Balseiro et al., 2019; Padala & Gujar, 2021),

while ensuring *incentive compatibility* (IC) (Hurwicz, 1973), i.e., that strategic agents cannot inflate their allocation by misrepresenting utilities.

This problem class breaks down along (i) the number of items to allocate, (ii) whether items are divisible or indivisible, (iii) the manner in which demand is communicated to the supplier (monetary or non-monetary preferences), (iv) the measure of fairness/social welfare. (v) whether the agents are truthful or strategic. Regarding (i): in the single-item multiple-agent case, optimal mechanism design has been resolved when monetary payments are made by the agents to the supplier (Myerson, 1981). Addressing the case of multiple bidders and items is the focus of this work, following Cai et al. (2012a;b); Yao (2014).

Regarding (ii), indivisible item settings exhibit fundamental bottlenecks due to combinatorial optimization underpinnings (Chevaleyre et al., 2006; Sönmez & Ünver, 2011). In this work, we study the divisible setting, which is applicable to, e.g., financial assets (Ko & Lin, 2008) and GPU hours (Ibrahim et al., 2016; Aguilera et al., 2014).

Regarding (iii): an agent requesting resources must indicate its interest in the form of a monetary bid or alternatively provide a utility function that represents its preferences. Numerous prior works on resource allocation consider auctions, where payments indicate interest in resources (Pavlov, 2011; Giannakopoulos & Koutsoupias, 2014; Dütting et al., 2023), yet many settings forbid payment (e.g. organ donation, food and necessity distribution by charity, allocation of GPU hours by an institution to its employees). Our work focuses on designing mechanisms without money (Dekel et al., 2010; Procaccia & Tennenholtz, 2013; Cole et al., 2013).

(iv) In the payment-free setting, measuring whether an allocation is fair among agents on a social level may be formalized through *proportional fairness* (**PF**) (Kelly, 1997). An allocation is said to achieve PF if any deviation from the allocation results in a non-positive change in the percentage utility gain summed over all agents. It is known (see Bertsimas et al. (2011)) that PF is achieved by maximizing Nash social welfare (NSW), the product of all agents' utilities, which we use as a quantifier of fairness/social welfare in this work.

What remains is (v) whether agents are truthful or strategic, i.e., they may misreport their preferences. In the later case, the mechanism needs to be designed so as not to provide incentive for the agents to misreport, i.e., to possess incentive compatibility (IC). The **PF mechanism**, which directly optimizes NSW of the allocation outcome assuming truthful reporting, is *not* incentive compatible. By contrast, several recent works study IC mechanism design without money (Dekel et al., 2010; Procaccia & Tennenholtz, 2013; Cole et al., 2013). Most germane to this discussion is the **Partial Allocation** (PA) mechanism (Cole et al., 2013) based on "money burning" (Hartline & Roughgarden, 2008), where the supplier intentionally withholds a proportion of the resources as an artificial payment. However, this mechanism is incentive compatible at significant cost to NSW.

In this work, we develop a *payment-free* mechanism that balances the competing criteria of fairness (in NSW sense) and IC. Our approach formulates the allocation problem as the one that maximizes NSW subject to a constraint on *exploitability*, which quantifies the maximum obtainable gain from misreporting and measures the deviation from exact IC. Similar trade-offs are considered in related resource allocation settings that require monetary payment (Dütting et al., 2023; Ivanov et al., 2022) and those not involving payments (Zeng et al., 2024), as well as in game-theoretic pricing models (Goktas & Greenwald, 2022). We detail our **main contributions** below.

- We develop a subgradient-based method for computing the exploitability of the PF mechanism. The PF mechanism acts as an important benchmark for welfare maximization in resource allocation. However, there exists no established method for computing the exploitability of the PF mechanism yet, making it difficult to compare against. This paper fills in the gap. As the PF mechanism is defined through an optimization program, our main innovation here is to derive the (sub)gradient of the output of the program with respect to the input, extending the results from the differentiable convex programming literature. This allows us to calculate the exploitability by iterative (sub)gradient ascent. We believe that the technique for differentiating through the PF mechanism may be of broader interest beyond mechanism design.

- We propose a novel neural network architecture, `RPF-Net`, that is highly effective for learning an approximately fair and IC allocation mechanism. `RPF-Net` can be interpreted from two perspectives. One is that it modifies the PF mechanism by adding a linear regularization designed to increase the robustness to potential misreports. The regularization is computed from the output of a neural network learned from data. From another perspective, `RPF-Net` can be viewed as standard neural network with a specific-purpose activation function tailored to the resource allocation problem structure. We derive the (sub)gradient of `RPF-Net`, which enables efficient end-to-end training.

- We provide a generalization bound of `RPF-Net` when it is trained on finite samples (Theorem 2). The result shows that the generalization error decays at rate $\mathcal{O}(L^{-1/2})$ where $L$ is the training sample size. In auction design with payment, RegretNet, the state-of-the-art learned mechanism proposed in Dütting et al. (2023), also achieves a $\mathcal{O}(L^{-1/2})$ rate, which we match in the non-payment setting. Our bound is derived by adapting and extending the techniques in Dütting et al. (2023) to the special activation function in `RPF-Net`.

- We provide a guarantee under distribution shift. The implication of this result is that we can train `RPF-Net` with samples from distribution $F'$ and expect them to perform well under distribution $F$, provided that the mismatch between $F$ and $F'$ is controlled.

- We experimentally validate `RPF-Net` across problem dimensions and distribution shift. The robustness to distribution shift, specifically, demonstrates the significance of our contribution by allowing the training data itself to be subjected to adversarial contamination.

## 1.1 Related Work

Our paper relates to the existing literature on resource allocation with and without monetary payments, as well as those that study incentive compatibility in the context of mechanism design and beyond. We discuss the most relevant works in these domains to give context to our contribution.

**Resource allocation without payment:** Guo & Conitzer (2010) considers the problem of allocating two divisible resources to two agents and show that any IC mechanism achieves at most 0.841 of the optimal social welfare in the worst case (defined as an overall utility of all agents). They further show that a linear increasing-price mechanism nearly achieves the lower bound. Han et al. (2011) obtains an analogous worst-case lower bound for a more general case of $n$ agents but does not provide a mechanism. Cole et al. (2013) proves that no IC mechanism can guarantee to every agent a fraction of their proportionally fair allocation greater than $\frac{n+1}{2n}$ as the number of resources becomes large, and proposes a partial allocation (PA) mechanism that allocates to each agent a fraction (at least $1/e$) of the corresponding PF allocation. These works assume that the supplier has no prior knowledge of the agents' preference (or its distribution), whereas our work considers the setting where (possibly inaccurate) historical samples of the agents' utility parameters are available for training. Also assuming the access to training samples, Zeng et al. (2024) is highly related to our work. Zeng et al. (2024) studies learning an approximately fair and IC mechanism named ExS-Net parameterized by a neural network and trains the network parameters on the same objective that we consider in this work. Our important distinction from ExS-Net lies in the activation function of the output layer: while the activation function in ExS-Net is composed of a simple softmax function and a synthetic agent which receives the portion of resources to be withheld, the activation function we propose leverages a convex optimization program. ExS-Net is an important baseline for comparison. In Section 6 we show that the proposed `RPF-Net` mechanism materially outperforms ExS-Net due to the innovation in the network architecture.

**Auction design/resource allocation with payment:** When payments from the agents to the supplier are allowed, Chawla et al. (2010); Roughgarden (2010); Brânzei et al. (2022) design sophisticated monetary transfer schemes that ensure truthful reporting. More recently, (Dütting et al., 2023; Ivanov et al., 2022) adopt a learning-based framework similar to the one in our paper. A neural-network-parameterized mechanism, `RegretNet`, is proposed in (Dütting et al., 2023), which determines the price of a resource with the aim of increasing supplier's revenue while guaranteeing the approximate truthfulness of the agents. Our work is partially inspired by `RegretNet`. Similar to our "differentiation through an optimization program" approach in spirit, Curry et al. (2022) learns approximately IC and revenue-maximizing auction mechanisms, designed

through a differentiable regularized linear program. Compared to `RegretNet`, the method in Curry et al. (2022) enlargens the solvable class of auctions.

**Approximate incentive compatibility in ML:** Dekel et al. (2010) consider regression learning where the global goal is to minimize average loss in the setting of strategic agents that might misreport their values over the input space. When payments are disallowed, they present a mechanism which is approximately IC and optimal for the special case of the buyer's utilities defined by the absolute loss. More recently, (Chen et al., 2020) focus on learning linear classifiers, when the training data comes in online manner from the strategic agents who can misreport the feature vectors, and propose an algorithm that exploits the geometry of the learner's action space. Ravindranath et al. (2021) is another highly related work that designs approximately IC and stable mechanisms for two-sided matching using concepts from differentiable economics.

**Organization.** The rest of the paper is structured as follows. In Section 2 we formulate the payment-free resource allocation problem and introduce the existing mechanisms. In Section 3, we present the proposed mechanism with the novel neural network architecture as well as the procedure to train the mechanism. In Section 4 we develop the methods for evaluating the exploitability of the PF mechanism and for back-propagating the gradient through `RPF-Net` during training. Section 5 provides guarantees on the performance of `RPF-Net` trained under finite and out-of-distribution samples. Simulations that illustrate the merits of the proposed mechanism are presented in Section 6. Finally, we conclude and reflect on the connection to the literature in Section 7.

## 2 Preliminaries & Problem Formulation

We study the problem of allocating a finite number $M$ of divisible resources to $N$ agents in the form of a vector $a \in \mathbb{R}^{NM}$, with $a_{i,m}$ the amount of resource $m$ allocated to agent $i$. There is a budget $b_m \geq 0$ on each resource $m = 1, \cdots, M$, and we denote $b = [b_1, \cdots, b_M]^\top \in \mathcal{B} \subseteq \mathbb{R}_+^M$. We assume that every agent has a (thresholded) linear additive utility – each additional unit of resource $m$ linearly increases the utility of agent $i$ by value $v_{i,m}$, up to the demand $x_{i,m}$. We represent the overall values and demands as $v_i = [v_{i,1}, \cdots, v_{i,M}]^\top$, $x_i = [x_{i,1}, \cdots, x_{i,M}]^\top$, and $v = [v_1^\top, \cdots, v_N^\top]^\top$, $x = [x_1^\top, \cdots, x_N^\top]^\top$. Given allocation $a \in \mathbb{R}_+^{NM}$, value $v \in \mathcal{V}$, and demand $x \in \mathcal{D}$, the utility function $u : \mathbb{R}_+^{NM} \times \mathcal{V} \times \mathcal{D} \to \mathbb{R}_+^N$ is agent-wise expressed as

$$u_i(a, v, x) \triangleq \sum_{m=1}^M v_{i,m} \min\{a_{i,m}, x_{i,m}\}, \tag{1}$$

with $\mathcal{V} \subseteq [\underline{v}, \overline{v}]^{NM}$, $\mathcal{D} \subseteq [\underline{d}, \overline{d}]^{NM}$ for some scalars $\underline{v}, \overline{v}, \underline{d}, \overline{d} > 0$.

The linear additive utility structure is widely considered and realistically models satisfaction in many practical problems (Bliem et al., 2016; Camacho et al., 2021), though the mechanism proposed in the paper may handle alternative utility functions. The supplier knows the functional form of the utility function and relies on each agent $i$ to report the parameters $v_i$ and $x_i$. It may be in the interest of the agents to report untruthfully.

Since the allocation needs to satisfy budget constraints and should not exceed what the agents request, valid allocations have to be contained in the set $\mathcal{A}_{b,x} = \{a \in \mathbb{R}^{NM} : 0 \leq a_{i,m} \leq x_{i,m}, \forall i, m, \sum_{i=1}^N a_{i,m} \leq b_i, \forall m\}$. A mechanism may incorporate non-negative agent weights $w \in \mathbb{R}_+^N$ to encode prioritization. Next, we formalize that a mechanism is a mapping from values, demands, budgets, and weights to a valid allocation.

**Definition 1** (Mechanism). *A mapping $f : \mathcal{V} \times \mathcal{D} \times \mathcal{B} \times \mathbb{R}_+^N \to \mathbb{R}_+^{NM}$ is a mechanism if $f(v, x, b, w) \in \mathcal{A}_{b,x}$ for all $v \in \mathcal{V}, x \in \mathcal{D}, b \in \mathcal{B}, w \in \mathbb{R}_+^N$.*

Agent weights $w$ may be determined solely by the supplier before the agents reveal their requests or can potentially be a function of the reported values and demands. Since agents may not directly alter weights, we suppress dependence on $w$ to write allocations as $f(v, x, b)$.

Two competing axes for characterizing the merit of a mechanism exist: social welfare and incentive compatibility (IC). Among the many social welfare metrics (see (Bertsimas et al., 2011)), we consider Nash social welfare (NSW), which balances fairness and efficiency.

**Definition 2** (Nash Social Welfare). *The NSW of a mechanism $f$ is*

$$\mathbf{NSW}(f, v, x, b) \triangleq \Pi_{i=1}^N u_i(f(v, x, b), v, x)^{w_i}, \tag{2}$$

Due to NSW as the product of agents' utilities, it may become numerically unstable as the number of agents scales up. In comparison, dealing with the logarithm NSW is usually more convenient

$$\mathbf{logNSW}(f, v, x, b) \triangleq \log(\mathbf{NSW}(f, v, x, b)).$$

The definition simplifies to $\mathbf{logNSW}(f, v, x, b) = w^\top \log u(f(v, x, b), v, x)$.

To quantify approximate incentive compatibility, we next introduce *exploitability* as the maximum positive unilateral deviation in an agent' utility when it misreports its preferences.

**Definition 3** (Exploitability). *Under mechanism $f$ and $v \in \mathcal{V}, x \in \mathcal{D}, b \in \mathcal{B}$, we define the exploitability at agent $i$ as its largest possible utility increase due to misreporting, given that the every agent $j \neq i$ reports $v_j, x_j$*

$$\mathbf{expl}_i(f, v, x, b) \triangleq \max_{v_i', x_i'} u_i\big(f((v_i', v_{-i}), (x_i', x_{-i}), b), v, x\big) - u_i\big(f(v, x, b), v, x\big). \tag{3}$$

A mechanism $f$ is **incentive compatible** (IC) if it satisfies $\mathbf{expl}_i(f, v, x, b) = 0$, for all $v, x, b$ and $i = 1, \cdots, N$.

Assuming that the agents truthfully report their preferences, the following mechanism directly seeks to optimize the (log) NSW and achieves the largest possible NSW by definition. The mechanism is usually referred to as the Proportional Fairness (**PF**) mechanism, as the solution satisfies the proportional fairness property: when switching to any other allocation the percentage utility changes across all agents sum up to a non-positive value (Caragiannis et al., 2019).

---

Mechanism: Proportional Fairness

$$f^{PF}(v, x, b) = \operatorname*{argmin}_{a \in \mathbb{R}^{NM}} \ -\sum_{i=1}^N w_i \log(a_i^\top v_i)$$

$$\text{s.t.} \quad 0 \leq a \leq x, \ \sum_{i=1}^N a_{i,m} \leq b_m, \quad \forall m. \tag{4}$$

---

With $D = \mathbf{1}_N^\top \otimes I_{M \times M} \in \mathbb{R}^{M \times NM}$, the last inequality of (4) can be expressed in the matrix form $Da \leq b$.

It is important to note that the PF mechanism is not IC, i.e., it has nonzero exploitability. This fact motivates us to develop ways to ensure it is approximately so by trading off NSW.

## 2.1 PF Mechanism Has a Non-Zero Exploitability

In this section, we illustrate the exploitability of the PF mechanism using a simple example. Consider a two-agent two-resource allocation problem, in which we choose $x = \mathbf{1}_{NM}$, $b = \mathbf{1}_M$, and $w = \mathbf{1}_N$. Both agents have a higher valuation for the first resource, with $v_1 = \{1, 1/2\}$ and $v_2 = \{1, 1/4\}$. When agent 2 reports truthfully, Figure 1 plots the utility of agent 1 [cf. (1)] as it varies the reported preference ratio $v_{1,2}/v_{1,1}$ from 0.1 to 3 (the true ratio is 0.5). With the dashed line indicating the utility of agent 1 under truthful reporting, under-reporting $v_{1,2}/v_{1,1}$ increases agent 1's utility up to 17%. By contrast, `RPF-Net` proposed in this work substantially reduces the exploitability.

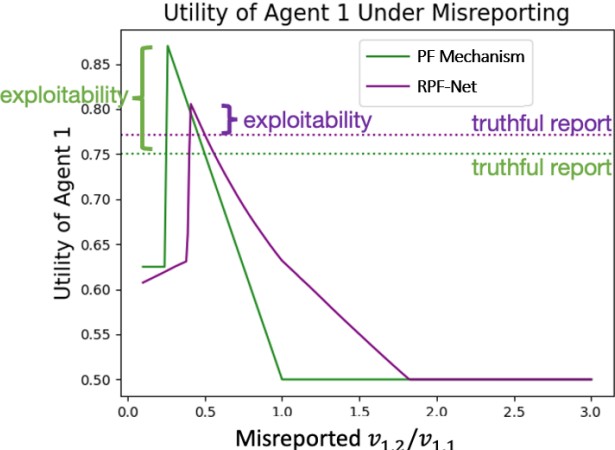

Figure 1: Exploitability of PF and learned `RPF-Net`.

The state-of-the-art IC mechanism in the non-payment setting is Partial Allocation (**PA**) (Cole et al., 2013), built upon the PF mechanism, which strategically withholds resources to ensure truthfulness[1]. Under the PA mechanism, each agent can only be

---

[1]Detailed presentation of the PA mechanism can be found in Appendix A.

guaranteed to at least receive a $1/e$ fraction of the resources that it would receive under the PF mechanism, meaning that there is resource waste and a significant reduction in NSW. In other words, the best known IC mechanism achieves a sub-optimal NSW, whereas the PF mechanism is optimal in NSW but incurs high exploitability. This dichotomy does not exist in the auction setting due to payments as an enforcement for truthful reporting. At least in the case $m = 1$, classic auction mechanisms including the VCG mechanism (Vickrey, 1961; Clarke, 1971; Groves, 1973) and Myerson mechanism (Myerson, 1981) can achieve IC and revenue maximization. However, in the payment-free setting, NSW maximization and IC are conflicting objectives that are provably not perfectly achievable at the same time (Hartline & Roughgarden, 2008; Cole et al., 2013).

## 2.2 Learning Approximate Payment-Free Mechanism

In this work, we are motivated to design a mechanism that approximately optimizes NSW and exploitability and strikes a desirable balance between them. Specifically, assuming that in a resource allocation problem the values and demands of the agents and the budgets on the resources follow a joint distribution $F$, we formalize our objective as maximizing the (log)NSW subject to a constraint on the exploitability for some $\epsilon \geq 0$.

$$
\begin{aligned}
\max_{\omega} \ & \mathbb{E}_{(v,x,b) \sim F}[\mathbf{logNSW}(f_\omega, v, x, b)] \\
\text{s.t. } & \mathbb{E}_{(v,x,b) \sim F}[\mathbf{expl}_i(f_\omega, v, x, b)] \leq \epsilon, \quad \forall i
\end{aligned}
\tag{5}
$$

We say a mechanism is $\epsilon$-incentive compatible over the distribution $F$ if it satisfies the constraints in (5).

PA and PF mechanisms are on the two ends of the trade-off between NSW and exploitability that are not optimal/feasible in the sense of (5). In this work, we propose parameterizing the mechanism with a neural network and learning the solution to (5) from data. This can be regarded as an adaptation of Dütting et al. (2023) to the payment-free setting. However, different from the auction setting, due to the impossibility result which states that IC cannot be achieved without "money burning" (i.e., resource withholding) (Hartline & Roughgarden, 2008), a regular neural network not equipped with the ability to withhold may fail to achieve low exploitability. In the following section, we propose a mechanism built on a novel neural network architecture specially designed for learning (5). We show later through numerical simulations that the architectural innovation is crucial – the mechanism with the proposed architecture performs substantially better than a learned mechanism parameterized by a regular neural network or ExS-Net (Zeng et al., 2024) which is the state-of-the-art learning-based solution.

# 3 Regularized Proportional Fairness Network

We develop a novel learning-based mechanism for resource allocation that can be regarded as a composition of a neural network (in this work we employ a feed-forward neural network of identical structure to parameterize both mechanisms, but in general any function approximation can be used) and a novel specific-purpose activation function. The mechanism, `RPF-Net` (**R**egularized **P**roportional **F**airness **Net**work), is a regularized variant of the PF mechanism that diverts allocation from what the agents would like to receive when they misreport to their largest advantage. A schematic representation of the mechanism is presented in Figure 2.

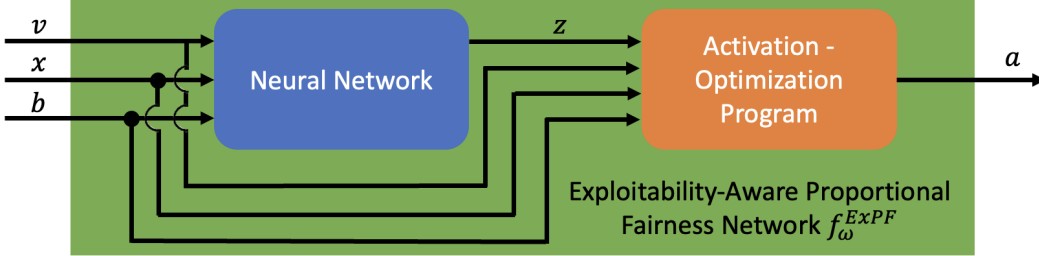

Figure 2: `RPF-Net` Pipeline.

### 3.1 Neural Network Design

The motivation behind `RPF-Net` is that we would like to reduce the agents' incentive to misreport through a linear regularization added to the NSW objective. We use a function approximation to track the allocation that an agent aims to receive when misreporting, and penalize such allocation to be produced.

Suppose that each agent $i$ has the perfect information about the mechanism and the parameters of others $(x_{-i}, v_{-i})$. The most adversarial misreport occurs when it maximizes its own utility under mechanism $f$, i.e., it would select $(\tilde{v}_i, \tilde{x}_i)$:

$$(\tilde{v}_i, \tilde{x}_i) = \mathrm{argmax}_{v_i', x_i'} u_i(f((v_i', v_{-i}), (x_i', x_{-i}), b), v, x). \tag{6}$$

Let $y_i = f_i((\tilde{v}_i, v_{-i}), (\tilde{x}_i, x_{-i}), b) \in \mathbb{R}^m$ denote the allocation to agent $i$ under the optimal misreport. The goal of agent $i$ when misreporting is to obtain allocation $y_i$. To ensure robustness to misreporting, the mechanism $f$ should avoid allocating $y_i$ to agent $i$, which could be enforced as a constraint:

$$\langle a_i, y_i \rangle \leq \delta, \ \forall i \tag{7}$$

for some constant $\delta > 0$. With $\delta$ selected sufficiently small, alignment between the allocation decision $a_i$ and the direction $y_i$ desired by the untruthful agent $i$ is penalized by the inner-product constraint, which prevents allocating $a_i = y_i$. In other words, the constraint (7) ensures that the allocation to an untruthful agent deviates from what it most desires, hence reducing the incentive to misreport. This line of reasoning is in the spirit of robust control (Zhou & Doyle, 1998; Borkar, 2002).

Denote the stacked vector of nominal allocations as $y = [y_1^\top, \cdots, y_N^\top]^\top \in \mathbb{R}^{NM}$. If $y$ were known, (4) with the additional inner-product constraint would be equivalent to

$$\mathrm{argmin}_{a \in \mathbb{R}^{NM}} \ -\sum_{i=1}^{N} w_i \log(a_i^\top v_i) + \sum_{i=1}^{N} \xi_i^\star \langle a_i, y_i \rangle$$
$$\text{s.t.} \quad 0 \leq a \leq x, \quad \sum_{i=1}^{N} a_{i,m} \leq b_m, \ \forall m \tag{8}$$

where $\xi_i^\star$ is the optimal dual variable associated with the inner-product constraint (7) for each $i$. Unfortunately, $y$ is difficult to compute as it encodes a notion of self-consistency – $y$ defines the mechanism (7) but is simultaneously the output of (7) under specific misreports. Due to difficulty of practical evaluation, we instead approximate $y$ from data.

We propose substituting $\xi_i^\star y_i$ by a parameterized function approximator. To be more precise, since the vector $y_i$ can be a function of values, demands, and budgets, we fit $z_\omega : \mathcal{V} \times \mathcal{D} \times \mathcal{B} \to \mathbb{R}^{NM}$ such that $[z_\omega(v, x, b)]_i$ tracks $\xi_i^\star y_i$ under reported values $v$, demands $x$, and budgets $b$. In this work, we take the function approximator to be a feed-forward neural network – $w$ represents the set of weights and biases while $z_w$ represents the neural network parameterized by $w$ as a mapping from the input values, demands, and budgets to an allocation outcome. Here $[z_\omega(v, x, b)]_i$ denotes the evaluation of the neural network restricted to the component associated with the allocation of agent $i$. Substituting $z_\omega$ into (8) leads to our first proposed approach, `RPF-Net`, whose associated mechanism [cf. Definition 1] is detailed below.

---

Mechanism: Exploitability-Aware PF

(1) Compute $z_\omega(v, x, b)$ as output of neural network
(2) Solve

$$f_\omega^{RPF}(v, x, b) = \mathrm{argmin}_{a \in \mathbb{R}^{NM}} \ -\sum_{i=1}^{N} w_i \log(a_i^\top v_i) + \langle a, z_\omega(v, x, b) \rangle$$
$$\text{s.t. } 0 \leq a \leq x, \ \sum_{i=1}^{N} a_{i,m} \leq b_m, \ \forall m. \tag{9}$$

---

Given a learned network parameter $\omega$, this procedure describes the operations to be performed in the inference phase. For any $\omega$, the output of the `RPF-Net` mechanism is always feasible, i.e., is non-negative, not exceeding

the demand, and within the budget constraint, by the definition of the optimization problem (9), which can be regarded as a special activation function operating on the output of the neural network $z_\omega(v, x, b)$. Note that for (9) to be a proper activation function we need to be able to back-propagate gradients through it. It is not obvious how we can back-propagate through (9) as the mapping from $z_\omega(v, x, b)$ to $f_\omega^{RPF}(v, x, b)$ is defined indirectly through an optimization problem. We address this important question in Section 4.3.

### 3.2 Training RPF-Net

While we would like to solve the objective (5) over distribution $F$, in practice, we often may not have access to $F$, but instead a training set of values, demand, and budgets $\{(v^l, x^l, b^l) \sim F\}_{l=1}^L$. We train the mechanisms with the finite dataset via empirical risk minimization (ERM) by forming the sample-averaged estimates of the expected NSW and exploitability.

$$\max_\omega \sum\nolimits_{l=1}^L \mathbf{logNSW}(f_\omega, v^l, x^l, b^l) \tag{10}$$
$$\text{s.t.} \sum\nolimits_{l=1}^L \mathbf{expl}_i(f_\omega, v^l, x^l, b^l) \leq \epsilon, \forall i$$

The sample-averaged performance of a mechanism approaches its expectation as the number of collected samples increases – the gap between them is known as **generalization error** (Shalev-Shwartz & Ben-David, 2014). In Sec. 5, we bound the generalization error by a sublinear function of batch size $L$ for both proposed mechanisms.

It is important to note that the learning objective does not require paired samples of $(v^l, x^l, b^l)$ and the ground-truth optimal allocation under $(v^l, x^l, b^l)$. We only need samples of valuations, demands, and budgets to learn the optimal parameter $w$ in an unsupervised manner.

To train the neural network, we optimize (10) with respect to $\omega$ by using a simple primal-dual gradient descent-ascent algorithm to find the saddle point of the Lagrangian. We present the training scheme in Alg. 1, where $\Pi_+$ denotes the projection of a scalar to the non-negative range. The dual variable $\gamma_i \in \mathbb{R}_+$ is associated with the $i_{\text{th}}$ exploitability constraint in (10).

---

**Algorithm 1:** Training `RPF-Net`

**Input:** Initial network parameter $\omega^{[0]}$, dual variables $\{\gamma_i^{[0]}\}_{i=1}^N$, training dataset $\{(v^l, x^l, b^l)\}_{l=1}^L$, batch size $s$, training iterations $K$, primal and dual learning rate $\alpha, \beta$

**Output:** Network parameter $\omega^{[K]}$.

**for** $k = 0, 1, \cdots, K-1$ **do**

  1) Randomly draw sample index set $\mathcal{S}^{[k]}$ with $|\mathcal{S}^{[k]}| = s$ and compute empirical logNSW and exploitability

$$\widehat{\mathbf{logNSW}}^{[k]} = \sum\nolimits_{l \in \mathcal{S}^{[k]}} \mathbf{logNSW}(f_{\omega^{[k]}}, v^l, x^l, b^l),$$
$$\widehat{\mathbf{expl}}_i^{[k]} = \sum\nolimits_{l \in \mathcal{S}^{[k]}} \mathbf{expl}_i(f_{\omega^{[k]}}, v^l, x^l, b^l), \quad \forall i$$

  2) Neural network parameter update:

$$\omega^{[k+1]} = \omega^{[k]} - \alpha \nabla_{\omega^{[k]}} \Big( \sum_i \gamma_i^{[k]} \widehat{\mathbf{expl}}_i^{[k]} - \widehat{\mathbf{logNSW}}^{[k]} \Big)$$

  3) Dual variable update:

$$\gamma_i^{[k+1]} = \Pi_+ \Big( \gamma_i^{[k]} + \beta \big( \widehat{\mathbf{expl}}_i^{[k]} - \epsilon \big) \Big), \quad \forall i$$

**end**

---

The `RPF-Net` mechanism is composed of a neural network and an optimization program. The latter can be interpreted as a special activation function on the network output. Passing $z$ through this activation in the

forward direction requires solving (9). Backward pass is practiced in the training stage, in which we need to back-propagate the training loss $\ell^{[k]} = \sum_i \gamma_i^{[k]} \widehat{\mathbf{expl}}_i^{[k]} - \widehat{\mathbf{logNSW}}^{[k]}$. As a critical step, we need to find the gradient $\frac{\partial \ell^{[k]}}{\partial z_{\omega^{[k]}}}$ given $\frac{\partial \ell^{[k]}}{\partial f_\omega^{RPF}}$. Computing such gradients is a challenging task, as the mapping from $z_\omega$ to the solution of (9) is defined implicitly through the optimization program. In the following section, we develop the technique that allows $\frac{\partial \ell^{[k]}}{\partial z_{\omega^{[k]}}}$ to be computed (relatively) efficiently. But still, the gradient computation requires inverting a square matrix of dimension $(3NM + M) \times (3NM + M)$. Both forward and backward operations can be increasingly expensive as the number of agents and resources scales up, which is a fact that we have to point out that may limit the scalability of `RPF-Net` in systems with computational constraints. Detailed complexity evaluation of `RPF-Net` is given in Section 6.4.

**Remark 1.** *We note that our work measures and optimizes exploitability only empirically and cannot guarantee strict exploitability constraint satisfaction, at least for two reasons. First, we cannot guarantee that gradient descent applied to the right hand side of Eq. (3) with respect to $v_i', x_i'$ finds the global maximizer, especially when the mechanism is a complicated mapping encoded by a neural network. Second, while we aim to optimize the objective (10), we cannot guarantee that the parameter $\omega$ obtained from training is a feasible solution. If the desired exploitability threshold $\epsilon$ is chosen too small, we sometimes observe constraint violation.*

## 4 PF Mechanism Exploitability Evaluation & RPF-Net Differentiation

Two main technical challenges are present at this point in training the proposed mechanism and evaluating its performance against benchmarks.

1. It is unclear how we may systematically calculate the exploitability of the PF mechanism, which is the fundamental baseline for any IC mechanism design. With a small number of resources, the exploitability can possibly be computed by exhaustively searching in the valuation and demand space, which we performed to generate Figure 1. However, as the system dimension scales up, an exhaustive parameter search quickly becomes computationally intractable. The main difficulty in computing $\mathbf{expl}_i(f^{PF}, v, x)$ lies in finding the optimal misreport $(\tilde{v}_i, \tilde{x}_i)$ as the solution to (6). It may be tempting to solve (6) with gradient ascent. However, as the mapping $f^{PF}$ is implicitly defined through an optimization problem (4), it is unclear how $\frac{\partial f^{PF}}{\partial v_i'}$ and $\frac{\partial f^{PF}}{\partial x_i'}$ (and potentially $\frac{\partial f^{PF}}{\partial w}$, since $w$ may be a function of $x_i', v_i'$) can be derived or even whether $f^{PF}$ is differentiable.

2. It is unclear how to back-propagate through `RPF-Net`. To train `RPF-Net` with gradient descent, we need to compute $\frac{\partial \ell}{\partial z_\omega}$ given $\frac{\partial \ell}{\partial f_\omega^{RPF}(v,x,b)}$ where $\ell$ is a downstream loss function calculated from $f_\omega^{RPF}(v, x, b)$.

We address both challenges in this section, by adapting and extending the techniques from differentiable convex programming (Amos & Kolter, 2017; Agrawal et al., 2019). As an important contribution of the work, we characterize the (sub)differentiability of $f^{PF}$ (Section 4.1) and develop an efficient method for calculating the (sub)gradients $\nabla_{v_i'} u_i, \nabla_{x_i'} u_i, \nabla_w u_i$ of (6) (Section 4.2). This allows iterative (sub)gradient ascent to be performed on the utility function $u_i$ to find a (locally) optimal solution of (6). This solution can then be leveraged to evaluate the exploitability of the PF mechanism according to (3). Building on a similar technique, we present an efficient method for differentiating through `RPF-Net` in Section 4.3.

The main property of $f^{PF}$ that we exploit to drive this innovation is the preservation of the KKT system of (4) at the optimal solution under differential changes to values and demands. We now present the detailed technical development.

### 4.1 Characterizing Sub-Differentiability of $f^{PF}$

Our goal is to determine how the differential changes in $v$, $x$, and $w$ affect the optimal solution of (4), which we denote by $a^\star(v, x, b) = f^{PF}(v, x, b)$ (or $a^\star$ for short when $v$, $x$, and $b$ are clear from the context). We

start by writing the KKT equations of (4) for stationarity, primal and dual feasibility, and complementary slackness

$$-\frac{w_i v_i}{v_i^\top a_i^\star} - \mu_i^\star + \nu_i^\star + \lambda^\star = 0, \quad \forall i \tag{11a}$$

$$\mu_{i,m}^\star a_{i,m}^\star = 0, \quad \forall i, m \tag{11b}$$

$$\nu_{i,m}^\star (a_{i,m}^\star - x_{i,m}) = 0, \quad \forall i, m \tag{11c}$$

$$\lambda_m^\star (Da^\star - b)_m = 0, \quad \forall m, \tag{11d}$$

where $\mu^\star, \nu^\star, \lambda^\star$ are the optimal dual solutions associated with constraints $a \geq 0$, $a \leq x$, and $Da \leq b$, respectively. As $a^\star$ satisfies $v_i^\top a_i^\star > 0$ for all $i$ (any allocation that makes $v_i^\top a_i^\star = 0$ for any $i$ blows up the objective of (4) to negative infinity and thus cannot be optimal), (11a) can be re-written as

$$(\mu_i^\star - \nu_i^\star - \lambda^\star) v_i^\top a_i^\star + w_i v_i = 0. \tag{12}$$

For (11) to hold when values, demands, and/or weights changes from $v, x, w$ to $v + dv, x + dx, w + dw$, the optimal primal and dual solutions need to adapt accordingly. We can describe the adaptation through the following system of equations on the differentials, which we obtain by differentiating (12) and (11b)-(11d).

$$(d\mu_i^\star - d\nu_i^\star - d\lambda^\star) v_i^\top a_i^\star + (\mu_i^\star - \nu_i^\star - \lambda^\star) dv_i^\top a_i^\star + (\mu_i^\star - \nu_i^\star - \lambda^\star) v_i^\top da_i^\star + w_i dv_i + dw_i v_i = 0, \quad \forall i,$$
$$d\mu_{i,m}^\star a_{i,m}^\star + \mu_{i,m}^\star da_{i,m}^\star = 0, \quad \forall i, m,$$
$$d\nu_{i,m}^\star (a_{i,m}^\star - x_{i,m}) + \nu_{i,m}^\star (da_{i,m}^\star - dx_{i,m}) = 0, \quad \forall i, m,$$
$$d\lambda_m^\star D_m a^\star + \lambda_m^\star D_m da^\star - d\lambda_m^\star b_m = 0, \quad \forall m.$$

We can equivalently express this system of equations in a compact matrix form

$$\boldsymbol{M} \left[ (da^\star)^\top, (d\mu^\star)^\top, (d\nu^\star)^\top, (d\lambda^\star)^\top \right]^\top = \boldsymbol{h}(dv, dx, dw), \tag{13}$$

where $\boldsymbol{M} \in \mathbb{R}^{(3NM+M) \times (3NM+M)}$ and $vh(dv, dx, dw) \in \mathbb{R}^{3NM+M}$ are

$$\boldsymbol{M} = \begin{bmatrix} \boldsymbol{M}_1 & \boldsymbol{M}_2 & -\boldsymbol{M}_2 & -\boldsymbol{M}_3 \\ \text{diag}(\mu^\star) & \text{diag}(a^\star) & 0 & 0 \\ \text{diag}(\nu^\star) & 0 & \text{diag}(a^\star - x) & 0 \\ \text{diag}(\lambda^\star) D & 0 & 0 & \text{diag}(Da^\star - b) \end{bmatrix} \quad \text{and} \quad \boldsymbol{h}(dv, dx, dw) = \begin{bmatrix} \boldsymbol{c}(dv, dw) \\ \boldsymbol{0} \\ \text{diag}(\nu^\star) dx \\ \boldsymbol{0} \end{bmatrix}.$$

The sub-matrices $\boldsymbol{M}_1, \boldsymbol{M}2 \in \mathbb{R}^{NM \times NM}, \boldsymbol{M}_3 \in \mathbb{R}^{NM \times M}$ and sub-vector $\boldsymbol{c} \in \mathbb{R}^{NM}$ are

$$\boldsymbol{M}_1 = \begin{bmatrix} (\mu_1^\star - \nu_1^\star - \lambda^\star) v_1^\top & \cdots & 0 \\ \vdots & \ddots & \vdots \\ 0 & \cdots & (\mu_N^\star - \nu_N^\star - \lambda^\star) v_N^\top \end{bmatrix}, \quad \boldsymbol{M}_2 = \begin{bmatrix} v_1^\top a_1^\star I_{M \times M} & \cdots & 0 \\ \vdots & \ddots & \vdots \\ 0 & \cdots & v_N^\top a_N^\star I_{M \times M} \end{bmatrix},$$

$$\boldsymbol{M}_3 = \begin{bmatrix} v_1^\top a_1^\star I_{M \times M} \\ \vdots \\ v_N^\top a_N^\star I_{M \times M} \end{bmatrix}, \quad \boldsymbol{c}(dv, dw) = - \begin{bmatrix} \big( (\mu_1^\star - \nu_1^\star - \lambda^\star)(a_1^\star)^\top + w_1 I_{M \times M} \big) dv_1 + v_1 dw_1 \\ \big( (\mu_2^\star - \nu_2^\star - \lambda^\star)(a_2^\star)^\top + w_2 I_{M \times M} \big) dv_2 + v_2 dw_2 \\ \vdots \\ \big( (\mu_N^\star - \nu_N^\star - \lambda^\star)(a_N^\star)^\top + w_N I_{M \times M} \big) dv_N + v_N dw_N \end{bmatrix}.$$

Under any differential changes $dv, dx, dw$, their impact on $da^\star$ can be derived by solving (13), which by definition gives the (sub)gradients when $dv, dx, dw$ are set to proper identity matrices/tensors.

**Example 4.1.** To compute $\frac{\partial a^\star}{\partial v_{1,1}}$, we evaluate how a unit change in $dv_{1,1}$ affects $da^\star$. If $\boldsymbol{M}$ is invertible, we solve

$$\left[ (da^\star)^\top, (d\mu^\star)^\top, (d\nu^\star)^\top, (d\lambda^\star)^\top \right]^\top = \boldsymbol{M}^{-1} \boldsymbol{h}(e_1, \boldsymbol{0}, \boldsymbol{0}) \tag{14}$$

and extract $da^\star$ from the solution, where $e_1 \in \mathbb{R}^{NM}$ denotes a vector with value 1 at the first entry and 0 otherwise. Under a general profile of values and demands, the matrix $\boldsymbol{M}$ need not be invertible. A singular matrix $\boldsymbol{M}$ leads to a non-differentiable mapping $f^{PF}$, but all solutions of

$$\boldsymbol{M}[(da^\star)^\top, (d\mu^\star)^\top, (d\nu^\star)^\top, (d\lambda^\star)^\top]^\top = \boldsymbol{h}(e_1, dx, dw)$$

are in the sub-differential of $a^\star$ at $v_{1,1}$.

**Theorem 1.** *Suppose that strict complementary slackness holds at solution $(a^\star, \mu^\star, \nu^\star, \lambda^\star)$ and that the demands are non-zero. When at least $NM - N$ inequality constraints in (4) hold as equalities at $a^\star$, the matrix $\boldsymbol{M}$ is invertible, and the mapping from $(v, x)$ to $a^\star$ is differentiable. Otherwise, the mapping is only sub-differentiable.*

Theorem 1 provides a sufficient condition for the differentiability of $f^{PF}$, and connects the differentiability to the number of tight constraints. The proof is presented in Appendix B. Similar characterizations have been established for quadratic programs and disciplined parameterized programs (Amos & Kolter, 2017; Agrawal et al., 2019). However, (4) does not fall (and cannot be re-formulated to fall) under either category. Hence, Theorem 1 generalizes these works to a broader category of convex programs; see Appendix E for further discussions.

## 4.2 Composing (Sub)Gradients

In principle, one can find $\frac{\partial a^\star}{\partial v}$ (and also $\frac{\partial a^\star}{\partial x}$, $\frac{\partial a^\star}{\partial w}$ with an identical approach) by repeatedly solving (14) for the unit change in each entry of $v$. However, doing so would require solving a large system of equations for every partial derivative or inverting $\boldsymbol{M}$ and storing its inverse, which is a costly operation. Fortunately, the following proposition shows that if $a^\star((v_{i'}, v_{-i}), (x_{i'}, x_{-i}), b)$ is used downstream to compute the utility $u_i(a^\star((v_{i'}, v_{-i}), (x_{i'}, x_{-i}), b), v, x)$ in (6), we can back-propagate the gradient $\nabla_{a^\star} u_i$ through the PF mechanism to obtain $\nabla_{v_{i'}} u_i$, $\nabla_{x_{i'}} u_i$, and $\nabla_w u_i$ by pre-computing and storing a matrix-vector product.

**Proposition 1.** *Under the same conditions as Theorem 1 and at least $NM - N$ inequality constraints of (4) are tight at $a^\star((v_{i'}, v_{-i}), (x_{i'}, x_{-i}), b)$, then given $\nabla_{a^\star} u_i$, we have*

$$\begin{aligned} \nabla_{v_{i'}} u_i &= \left((\nu_i^\star + \lambda^\star - \mu_i^\star)(a_i^\star)^\top - w_i I_{M \times M}\right) g_{a, iM:(i+1)M}, \\ \nabla_{x_{i'}} u_i &= \mathrm{diag}(\nu^\star) g_\nu, \quad \nabla_{w_i} u_i = v_i^\top g_{a, iM:(i+1)M}, \end{aligned} \tag{15}$$

$$\text{with } \boldsymbol{M}^\top \left[g_a^\top, g_\mu^\top, g_\nu^\top, g_\lambda^\top\right]^\top = \left[(\nabla_{a^\star} u_i)^\top, \boldsymbol{0}^\top, \boldsymbol{0}^\top, \boldsymbol{0}^\top\right]^\top. \tag{16}$$

If the assumptions in Prop. 1 do not hold, $\boldsymbol{M}$ is singular, whereby (15) defines subgradients of $u_i$, with $g_a$ and $g_\nu$ as any solution of (16). In practice, we use the one with minimal $\ell_2$ norm. These results enable performing (sub)gradient ascent to compute exploitability of the PF mechanism.

## 4.3 RPF-Net Differentiation

We can repeat the steps in Sections 4.1 and 4.2 on the modified objective (9) and derive the gradient of a loss function $\ell$, computed from $\partial f_\omega^{RPF}(v, x, b)$, with respect to $z_\omega$ when we are given $\frac{\partial \ell}{\partial f_\omega^{RPF}(v, x, b)}$[2]. We need to solve the system of equations

$$(\boldsymbol{M}')^\top \left[g_a^\top, g_\mu^\top, g_\nu^\top, g_\lambda^\top\right]^\top = \left[\left(\frac{\partial \ell}{\partial f_\omega^{RPF}(v, x, b)}\right)^\top, \boldsymbol{0}^\top, \boldsymbol{0}^\top, \boldsymbol{0}^\top\right]^\top$$

where the matrix $\boldsymbol{M}' \in \mathbb{R}^{(3NM+M) \times (3NM+M)}$, defined in the appendix, shares common entries as $\boldsymbol{M}$. Then, with $a^\star = f_\omega^{RPF}(v, x, b)$, we have

$$\left(\frac{\partial \ell}{\partial [z_\omega(v, x, b)]_i}\right) = v_i^\top a_i^\star g_{a, iM:(i+1)M}.$$

---

[2]The detailed derivation is deferred to Appendix D.

To evaluate the exploitability of the `RPF-Net` mechanism using gradient descent as outlined in Section 4, we also need to find $\frac{\partial \ell}{\partial v_i}$, $\frac{\partial \ell}{\partial x}$, and possibly $\frac{\partial \ell}{\partial w_i}$. Their expressions are given as follows

$$\frac{\partial \ell}{\partial v_i} = \Big( -w_i I_{M \times M} - (\mu_i^\star - \nu_i^\star - \lambda^\star - z_i)(a_i^\star)^\top \Big) g_{a,iM:(i+1)M}, \quad \frac{\partial \ell}{\partial x} = \mathrm{diag}(\nu^\star) g_\nu, \quad \frac{\partial \ell}{\partial w_i} = v_i^\top g_{a,iM:(i+1)M},$$

where $\mu^\star, \nu^\star, \lambda^\star$ are the optimal dual solutions of (9) associated with constraints $a \geq 0$, $a \leq x$, and $Da \leq b$.

## 5 Theoretical Guarantees

In this section, we establish the convergence of the learned mechanism. First, we bound the sub-sampling error by a sublinear function of the batch size, which matches recent rates for auctions (Dütting et al., 2023), and ensures the objective of (10) converges to (5) with enough data. Next, we show that the learned mechanism is robust under distribution mismatch, i.e., when we train (5) on a distribution $F'$ and evaluate on distinct distribution $F$, the learned mechanism achieves low exploitability with performance determined by distributional distance between $F$ and $F'$.

### 5.1 Generalization Bounds

For mechanism $f$, we define the generalization errors with $L$ samples as

$$\varepsilon_{\textbf{logNSW}}(f, L) = -\mathbb{E}_{(v,x,b) \sim F}[\textbf{logNSW}(f, v, x, b)] + \sum_{l=1}^{L} \textbf{logNSW}(f, v^l, x^l, b^l),$$

$$\varepsilon_{\textbf{exp},i}(f, L) = \mathbb{E}_{(v,x,b) \sim F}[\textbf{expl}_i(f, v, x, b)] - \sum_{l=1}^{L} \textbf{expl}_i(f, v^l, x^l, b^l).$$

With a slight abuse of notation, let $u^\omega(v, x, b) := u(f^\omega(v, x, b), v, x)$ and let $y \mapsto (v, x, b)$.

**Assumption 1.** *For each agent $i$, assume that $\frac{1}{\psi} \leq u_i^\omega(y) \leq \psi$, $\forall \, y \in F$ and some $\psi > 1$, and $v_i(S) \leq 1$, $\forall \, i$ and all subsets of resources.*

An implication of this assumption is that the activation function of `RPF-Net` is $\Phi-$Lipschitz for some $\Phi > 0$.

**Theorem 2** (Generalization Bound)**.** *Consider `RPF-Net` parameterized by a neural network with $R$ hidden layers with ReLU activation and $K$ nodes per hidden layer. Let $d$ denote the total parameters with the vector of all model parameters $\|\omega\|_1 \leq \Omega$. Under Assumption 1, the following holds with probability at least $1 - \delta$*

$$\max\{\varepsilon_{\textbf{logNSW}}(f_\omega^{RPF}, L), \varepsilon_{\textbf{exp},i}(f_\omega^{RPF}, L)\} \leq \mathcal{O}\Big(\psi N \frac{\sqrt{Rd \log(LN\Omega\Phi \max\{K, MN\})}}{L} + N\sqrt{\frac{\log(1/\delta)}{L}}\Big).$$

The result provides that the generalization error decays at rate $\mathcal{O}(L^{-1/2})$ where $L$ is the training sample size. In auction design with payment, RegretNet, the state-of-the-art learned mechanism proposed in Dütting et al. (2023), also achieves a $\mathcal{O}(L^{-1/2})$ rate, which we match in the non-payment setting. The proof of Theorem 2 is inspired by Dütting et al. (2023), but extended to handle: i) the fairness objective (NSW) in the absence of payment; ii) the special activation function in `RPF-Net`. The arguments in the proof can be used to obtain a similar order on the generalization error for the ExS-Net architecture of Zeng et al. (2024) as well.

### 5.2 Robustness to Distribution Mismatch

Let $\omega^\star(F)$ denote an optimizer of (10) under samples $\{(v^l, x^l, b^l)\}_l$ drawn from the distribution $F$. Next, we establish that the performance of $f_{\omega^\star(F')}$ on samples from distribution $F$ when trained on samples from a different distribution $F'$ in terms of the worst-case exploitability is controlled by the degree of mismatch.

**Theorem 3.** *Suppose the mechanism $f_\omega^{RPF}$ with parameter $\omega$ is $\epsilon$-incentive compatible over distribution $F'$. We have for any $i = 1, \cdots, N$*

$$E_{(v,x,b) \sim F}[\textbf{expl}_i(f, v, x, b)] \leq \epsilon + 2M \, \overline{v} \, \overline{x} \, d_{TV}(F, F'),$$

*where $d_{TV}$ denotes the total variation (TV) distance.*

As an implication of Theorem 3, we can train `RPF-Net` with samples from distribution $F'$ and expect them to perform well under distribution $F$, provided that the mismatch between $F$ and $F'$ is not arbitrarily large. Later in Sec. 6.2, we will verify this result through numerical simulations. The proof of Theorem 3 is presented in Appendix G.

## 6 Experiments

We numerically evaluate `RPF-Net` i) as the number of agents and resources changes; and ii) when the mechanisms are tested on a distribution different from that observed during training. We also visualize the decision boundary of `RPF-Net` in contrast to the PF mechanism to provide more insight on how `RPF-Net` deviates from the PF mechanism which it is designed to approximate and enhance. The final subsection discusses the computational time required for training and performing inference with `RPF-Net`.

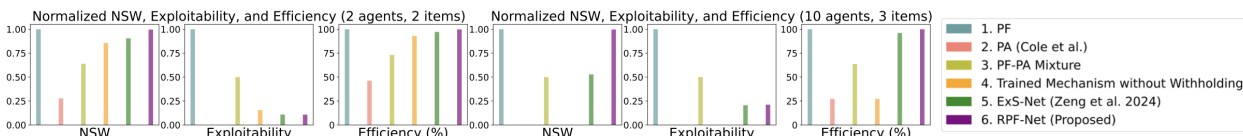

Figure 3: Mechanism performance in 2x2 and 10x3 systems (normalized with respect to PF mechanism)

**Evaluation Metrics:** Mechanisms are evaluated on NSW (2), exploitability

$$\mathbf{expl}(f, v, x, b) = (1/N) \sum_{i=1}^{N} \mathbf{expl}_i(f, v, x, b),$$

and efficiency

$$\mathbf{efficiency}(f, v, x, b) = \frac{\sum_{m=1}^{M} \sum_{i=1}^{N} f_{i,m}(v, x, b)}{\sum_{m=1}^{M} b_m}.$$

Efficiency quantifies the averaged utilization of resources. Mechanisms should preferably have a high efficiency, although efficiency is merely a byproduct of NSW in the training objective. As the total demand for a resource may be smaller than the budget, the maximum efficiency can be smaller than 1. The PF mechanism is fully efficient (i.e., all available resources are allocated if the demand is sufficiently large), and hence serves as a benchmark.

**Baselines:** Besides the PF mechanism introduced in (4), we evaluate against the PA mechanism (Cole et al., 2013), denoted by $f^{PA}$, which achieves state-of-the-art NSW among IC mechanisms. A probabilistic mixture of PA and PF provides a strong trade-off between NSW and exploitability. Given $\rho \in [0, 1]$, we consider a new mechanism $f^{\text{mixture}}$:

$$r \sim \text{Bern}(\rho), \ f^{\text{mixture}}(v, x, b) = \begin{cases} f^{PF}(v, x, b), & \text{if } r = 1 \\ f^{PA}(v, x, b), & \text{if } r = 0 \end{cases}$$

Varying $\rho$ between $[0, 1]$ linearly interpolates PF and PA in expectation. We set $\rho = 1/2$ in the experiments.

Another important baseline is the ExS-Net mechanism proposed in Zeng et al. (2024) which shares the same training objective as `RPF-Net` and is different in the neural network activation function.

We also compare against a standard neural network, which has the same architecture as `RPF-Net` and ExS-Net except that the activation function of its last layer is a standard softmax function and is trained on the same objective for the same amount of iterations. Any performance gain of `RPF-Net` over ExS-Net and this standard network can be attributed to the architectural innovation.

**Data Generation:** In all experiments, the true values and demands follow uniform and Bernoulli uniform distributions, respectively, within the range $[0.1, 1]$. Specifically, we generate the test samples according to

$$v_{i,m} \sim \text{Unif}(0.1, 1) \,, \widetilde{x}_{i,m} \sim \text{Unif}(0.1, 1) \,, \tag{17}$$

$$\widehat{x}_{i,m} \sim \text{Bern}(0.5) \, , x_{i,m} = \widetilde{x}_{i,m}\widehat{x}_{i,m}.$$

Unless noted otherwise in Sec. 6.1, the budget for each resource is set to $\frac{N}{2}$, where $N$ is the number of agents. This creates moderate competition for the resources in expectation. Training data is sampled from the same distributions as test data according to (17) except in Sec. 6.2 which studies distribution mismatch. All agent weights are set to 1.

## 6.1 Varying System Parameters

We test the proposed mechanism as the problem dimension varies. We start with two small-scale problems with 2 agents, 2 resources, and 10 agents, 3 resources. The 2x2 system is the smallest non-trivial case, and the 10x3 system is the largest considered in a line of recent works (Dütting et al., 2023; Ivanov et al., 2022) on auction design with payment. All reported numbers are normalized by that of the PF mechanism. As shown in Figure 3, `RPF-Net` achieves an advantageous trade-off between PF and PA: it consistently reduces the exploitability of PF by over at least 80% while achieving similar NSW. Compared with PA, `RPF-Net` improves the efficiency and NSW. For larger numbers of agents, the NSW of PA mechanism decreases drastically being the product of agents' utilities, while `RPF-Net` still exhibits a favorable NSW. In addition, `RPF-Net` outperforms the interpolated mixture of PF and PA and ExS-Net under all three metrics. The performance of the standard neural network is unstable (mechanism 4 in orange) – while it performs well in the 2x2 system, it fails to achieve a meaningful NSW with 10 agents present.

## 6.2 Distribution Mismatch

It can be difficult in practical problems to know and sample from the true distribution of values, demands, and budgets in the test set. In this section, we show that `RPF-Net` is still effective when the distribution on which they are expected to perform is slightly different from the training distribution. Such distribution mismatch may occur due to measurement error or more pernicious sources such as strategically misrepresenting preferences.

Recall that $F$ denotes the true distribution of utility function parameters and $F'$ the distribution of the training samples. We denote by $\omega^{\star}(F')$ the optimal solution to (10) under samples $\{(v^l, x^l, b^l) \sim F'\}$. The performance of $\omega^{\star}(F')$ under the true distribution $F$, measured by $\mathbb{E}_F[\textbf{NSW}(\omega^{\star}(F'), v, x, b)]$ and $\mathbb{E}_F[\textbf{expl}(\omega^{\star}(F'), v, x, b)]$, can in theory be sub-optimal by a factor of $d_{TV}(F, F')$, which is non-ideal under a large discrepancy between $F$ and $F'$. Nonetheless, empirically we observe robustness: we study two sources of distribution mismatch and see in the experiments that $\mathbb{E}_F[\textbf{NSW}(\omega^{\star}(F'), v, x, b)]$ and $\mathbb{E}_F[\textbf{expl}(\omega^{\star}(F'), v, x, b)]$ closely match $\mathbb{E}_F[\textbf{NSW}(\omega^{\star}(F), v, x, b)]$ and $\mathbb{E}_F[\textbf{expl}(\omega^{\star}(F), v, x, b)]$.

**Randomly perturbed training samples.** In a $2 \times 2$ system we suppose that the distribution $F$ follows (17) while each training sample $\{(v^l, x^l)\}_l$ is generated according to

$$
\begin{aligned}
&\bar{v}_{i,m}^l \sim \text{Unif}(0.1, 1), \ \bar{x}_{i,m}^l \sim \text{Unif}(0.1, 1). \\
&\tilde{v}_{i,m}^l \sim \text{Cauchy}(0, 0.01), \ \tilde{x}_{i,m}^l \sim \text{Cauchy}(0, 0.01), \\
&v_{i,m}^l = [\bar{v}_{i,m}^l + \tilde{v}_{i,m}^l]_{[0.1,1]}, \\
&\widehat{x}_{i,m} \sim \text{Bern}(0.5), \ x_{i,m}^l = \widehat{x}_{i,m}[\bar{x}_{i,m}^l + \tilde{x}_{i,m}^l]_{[0.1,1]}.
\end{aligned}
\tag{18}
$$

Here $[\cdot]_{[a,b]}$ for scalar $a, b$ denotes the element-wise projection to the interval $[a, b]$. In other words, $F'$ has a perturbed uniform distribution. We would like to draw the perturbation from a heavy-tailed distribution to increase the likelihood of generating extreme values. The Cauchy distribution has been used to explain extreme events like Flash Crash (Parker, 2016) and to describe price fluctuations (Casault et al., 2011), which motivates our selection.

**Adversarially generated training samples.** Again, we consider a $2 \times 2$ system where the true distribution $F$ is described in (17). Suppose that the training dataset is composed of historical valuations and demands collected through past interactions of the agents with the supplier. If the agents believe that the supplier runs the PF mechanism in these past interactions, they may have reported strategically to "trick" the PF

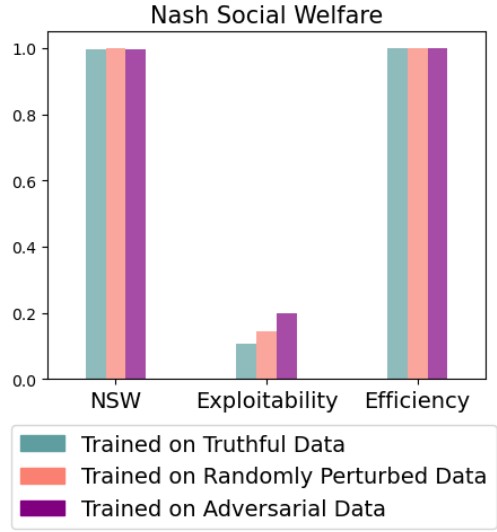

Figure 4: Performance of `RPF-Net` trained under data containing untruthful report. Numbers normalized by NSW/exploitability/efficiency of PF mechanism.

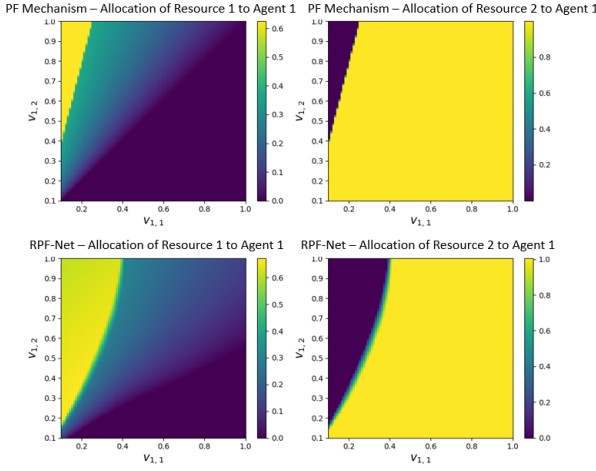

Figure 5: Allocations from PF (top) and `RPF-Net` (bottom).

mechanism. The training samples $\{(v^l, x^l)\}_l$ will then take the following form:

$$
\begin{aligned}
&\bar{v}_{i,m}^l \sim \text{Unif}(0.1, 1), \quad \bar{x}_{i,m}^l \sim \text{Unif}(0.1, 1). \\
&v_i^l, x_i^l = \arg\max_{v_i', x_i'} u_i(f^{PF}((v_i', \bar{v}_{-i}^l), (x_i', \bar{x}_{-i}^l), b), \bar{v}^l, \bar{x}^l).
\end{aligned}
\tag{19}
$$

In Table 1 and Figure 4 we present the NSW, exploitability, and efficiency of `RPF-Net` trained under contaminated samples (18) and (19). We observe that under contaminated training samples the metrics of `RPF-Net` closely track those under truthful training data. Notably, the NSW and efficiency are almost unaffected, while the exploitability only mildly increases. The proposed mechanism still achieves near-optimal NSW and efficiency while maintaining a low exploitability (not exceeding 20% of that of the PF mechanism). This demonstrates the robustness of the proposed mechanism to mismatch between training and inference distribution and supports the theoretical results in Sec. 5.2.

| Mechanism | NSW | Exploitability | Efficiency (%) |
|---|---|---|---|
| Proportional Fairness Mechanism | 8.00e-2±1.5e-2 | 3.70e-3±1.2e-3 | 56.6±3.9 |
| Partial Allocation Mechanism | 2.22e-2±2.1e-3 | 0±0 | 26.2±2.5 |
| PF-PA Mixture | 5.11e-2±6.6e-3 | 1.85e-3±6.2e-4 | 41.4±3.6 |
| `RPF-Net` (Trained on Truthful Data) | 7.97e-2±1.4e-2 | 3.99e-4±1.2e-4 | 56.5±4.0 |
| `RPF-Net` (Trained on Randomly Perturbed Data) | 7.99e-2±1.5e-2 | 5.3e-4±2.4e-4 | 56.5±4.0 |
| `RPF-Net` (Trained on Adversarial Data) | 7.98e-2±1.5e-2 | 7.4e-4±8.0e-5 | 56.5±3.9 |

Table 1: Performance of `RPF-Net` trained under data containing untruthful report. Mean and standard deviation reported.

## 6.3 Visualizing Allocation Under RPF-Net

To shed more light on `RPF-Net`, we visualize and compare the allocation vectors generated by PF and well-trained `RPF-Net` on a 2-agent 2-resource allocation problem. Under a specific profile of preferences and

with the reporting of agent 2 fixed, Figure 5 plots the allocation to agent 1 as $v_{1,1}$ and $v_{1,2}$ vary at the same time. PF has clear and sharp decision boundaries. As an interesting observation, the learned `RPF-Net` mechanism closely tracks the boundary of PF but determines allocations in a smoother way.

### 6.4 Complexity of Proposed Mechanism in Training and Inference Phase

In this section, we provide more light on the amount of computation required to train `RPF-Net` and use it for inference. For this purpose, it is worth comparing with the existing mechanism ExS-Net proposed in Zeng et al. (2024) which considers the same training objective as ours but parameterizes the mechanism with a much simpler regular neural network architecture. The main difference between `RPF-Net` and ExS-Net (Zeng et al., 2024) lies in the activation function. In the case of ExS-Net, the activation involves standard scalar/vector operation and a softmax function. Both forward and backward passes through the activation can be completed in time $\mathcal{O}(NM)$, which means that ExS-Net is fast in both training and inference phases. For `RPF-Net`, forward pass requires solving the convex optimization program (9). We use an interior-point solver for this program, which is guaranteed to converge within polynomial time, i.e., the time to obtain an solution up to precision $\varepsilon$ is no more than some polynomial function of $\varepsilon$, $N$, and $M$. However, the exact complexity is unknown but should be expected to be worse than $\mathcal{O}((NM)^3)$ (Renegar, 1988) (as $\mathcal{O}((NM)^3)$ is the time it would take for the interior-point method to converge if (9) were a linear program). In the backward pass, `RPF-Net` needs to invert solve a system of equations of dimension $(3NM + M) \times (3NM + M)$, which requires $\mathcal{O}((NM)^3)$ computation. To summarize, we organize the complexity results in Table 2.

We also show in Table 2 the training and inference time on a ten-agent three-resource allocation problem. Note that both PF and PA are hand-designed mechanisms that do not require training, but are time-consuming during inference since they solve optimization programs. The amount of computation required by `RPF-Net` during inference is on the same order as that of PF and PA mechanisms.

| Mechanism | Training Time (Theory) | Inference Time (Theory) | Training Time (Simulation) | Inference Time (Simulation) |
|---|---|---|---|---|
| Proportional Fairness | 0 | At least $\mathcal{O}((MN)^3)$ | 0 | 40.1 |
| Partial Allocation | 0 | At least $\mathcal{O}((MN)^3)$ | 0 | 310.0 |
| ExS-Net | $\mathcal{O}(MN)$ | $\mathcal{O}(MN)$ | 1 | 1 |
| `RPF-Net` (Proposed) | At least $\mathcal{O}((MN)^3)$ | At least $\mathcal{O}((MN)^3)$ | 380.2 | 114.5 |

Table 2: Training and inference time for a ten-agent three-resource problem with samples generated according to (17). Reported training time and inference time of all mechanisms are normalized with respect to those of ExS-Net.

## 7 Conclusion

This paper studied the important problem of fair and incentive compatible resource allocation without the use of monetary payment. Identifying that hand-designing mechanisms that achieve the exact IC and maximum NSW is impossible, we considered learning an approximate mechanism that desirably trades off the two competing objectives. As a key contribution, we innovated the neural network architecture used to parameterize the mechanism – we proposed `RPF-Net`, a learned variant of the standard PF mechanism with an optimization-based activation function in the output layer. We established a way of differentiating through the activation function to enable efficient training. On the theoretical side, we showed that the proposed mechanism is consistent (learns better with more training data) and is robust to distributional changes in the data, two properties that make the mechanism useful in practice. We showed through numerical simulations that `RPF-Net` outperforms the existing methods on a range of evaluation criteria. In particular, compared to the state-of-the-art architecture (Zeng et al., 2024), `RPF-Net` significantly increases the NSW without incurring a higher exploitability, thereby improving the Pareto frontier.

It is worth noting the limitation of `RPF-Net` in its computational complexity. The optimization-based activation function requires significant amount of computation to evaluate in the forward direction and to

differentiate through in the backward direction. Enhancing the computational complexity of the architecture while not compromising performance is an important future work.

## Disclaimer

This paper was prepared for informational purposes ["in part" if the work is collaborative with external partners] by the Artificial Intelligence Research group of JPMorgan Chase & Co. and its affiliates ("JP Morgan") and is not a product of the Research Department of JP Morgan. JP Morgan makes no representation and warranty whatsoever and disclaims all liability, for the completeness, accuracy or reliability of the information contained herein. This document is not intended as investment research or investment advice, or a recommendation, offer or solicitation for the purchase or sale of any security, financial instrument, financial product or service, or to be used in any way for evaluating the merits of participating in any transaction, and shall not constitute a solicitation under any jurisdiction or to any person, if such solicitation under such jurisdiction or to such person would be unlawful.

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

## Contents

## A   Partial Allocation Mechanism

The partial allocation mechanism, proposed in Cole et al. (2013), is built upon the PF mechanism and withholds resources according to an externality ratio. Specifically, given reported valuations $v \in \mathbb{R}^{N \times M}$ and demands $d \in \mathbb{R}^{N \times M}$ across all agents and resources and budgets $b \in \mathbb{R}^M$, let $a^\star \triangleq f^{PF}(v, x, b)$ denote the allocation made by the PF mechanism.

Let $x^{-i} \in \mathbb{R}^{N \times M}$ denote a modified demand matrix where

$$x_j^{-i} = \begin{cases} x_j \in \mathbb{R}^M, & i \neq j \\ 0 \in \mathbb{R}^M, & i = j \end{cases}$$

For each agent $i$, we calculate

$$a^{-i,\star} \triangleq f^{PF}(v, x^{-i}, b),$$

which is the PF allocation outcome that would arise in the absence of agent $i$. We further compute the externality ratio $r_i \in \mathbb{R}_+$ for agent $i$ as

$$r_i = \left( \frac{\prod_{j \neq i} u_j(a^\star, v, x)^{w_j}}{\prod_{j \neq i} u_j(a^{-i,\star}, v, x)^{w_j}} \right)^{1/w_i},$$

where $u_i$ is the utility function defined in (1).

The PA mechanism allocates to agent $i$ according to the following rule

$$f_i^{PA}(v, x, b) = r_i a_i^\star.$$

It is easy to verify that $r_i \in [0, 1]$. Therefore, the PA mechanism is always feasible. Cole et al. (2013)[Theorem 3.2] proves that it is also perfectly incentive compatible for any $v, x, b$.

## B   Proof of Theorem 1

To show the invertibility of $\boldsymbol{M}$ when at least $NM - N$ constraints are tight, it is equivalent to show that the following system of equations has a unique solution $[da^\top, d\mu^\top, d\nu^\top, d\lambda^\top]^\top$ for any vector $[\alpha^\top, \beta^\top, \gamma^\top, \zeta^\top]^\top$

$$\boldsymbol{M} \begin{bmatrix} da \\ d\mu \\ d\nu \\ d\lambda \end{bmatrix} = \begin{bmatrix} \alpha \\ \beta \\ \gamma \\ \zeta \end{bmatrix}.$$

We define $\mathcal{I}_\mu \triangleq \{(i,m) \in [1, \cdots, N] \times [1, \cdots, M] \mid \mu_{i,m}^\star > 0\} = \{(i,m) \in [1, \cdots, N] \times [1, \cdots, M] \mid a_{i,m}^\star = 0\}$, where the second equality is due to strict complementary slackness. Similarly, we define $\mathcal{I}_\nu \triangleq \{(i,m) \in [1, \cdots, N] \times [1, \cdots, M] \mid \nu_{i,m}^\star > 0\} = \{(i,m) \in [1, \cdots, N] \times [1, \cdots, M] \mid a_{i,m}^\star = x_{i,m}\}$, and $I_\lambda = \{m \in [1, \cdots, M] : \lambda_m > 0\} = \{m \in [1, \cdots, M] : \sum_i a_{i,m}^\star = b_m\}$. We use $\mathcal{I}_\mu^c, \mathcal{I}_\nu^c, \mathcal{I}_\lambda^c$ to denote their complement sets.

Note that for any $(i,m) \notin \mathcal{I}_\mu$, its corresponding row in the second row block of M only has one non-zero entry $a_{i,m}^\star$ in the second column block of $M$. This implies the solution $d\mu_{i,m} = \frac{\beta_{i,m}}{a_{i,m}^\star}$. Similarly, for any $(i,m) \notin \mathcal{I}_\nu$

and $m \notin \mathcal{I}_\lambda$, we are able to immediately write the solutions $d\nu_{i,m}$ and $d\lambda_m$ and remove them from the system of equations. Eventually, the system to be solved reduces to

$$
\begin{bmatrix}
\boldsymbol{M}_1 & (\boldsymbol{M}_2)_{:,\mathcal{I}_\mu} & -(\boldsymbol{M}_2)_{:,\mathcal{I}_\nu} & -(\boldsymbol{M}_3)_{:,\mathcal{I}_\lambda} \\
\mathrm{diag}(\mu^\star_{\mathcal{I}_\mu}) & \mathrm{diag}(a^\star_{\mathcal{I}_\mu}) & 0 & 0 \\
\mathrm{diag}(\nu^\star_{\mathcal{I}_\nu}) & 0 & \mathrm{diag}((a^\star - x)_{\mathcal{I}_\mu}) & 0 \\
\mathrm{diag}(\lambda^\star_{\mathcal{I}_\lambda})D_{\mathcal{I}_\lambda} & 0 & 0 & \mathrm{diag}((Da^\star - b)_{\mathcal{I}_\lambda})
\end{bmatrix}
\begin{bmatrix}
da \\
d\mu_{\mathcal{I}_\mu} \\
d\nu_{\mathcal{I}_\nu} \\
d\lambda_{\mathcal{I}_\lambda}
\end{bmatrix}
$$

$$
=
\begin{bmatrix}
\alpha - (\boldsymbol{M}_2)_{:,\mathcal{I}^c_\mu}d\mu_{\mathcal{I}^c_\mu} + (\boldsymbol{M}_2)_{:,\mathcal{I}^c_\nu}d\mu_{\mathcal{I}^c_\nu} + (\boldsymbol{M}_3)_{:,\mathcal{I}^c_\lambda}d\mu_{\mathcal{I}^c_\lambda} \\
\beta_{\mathcal{I}_\mu} \\
\gamma_{\mathcal{I}_\nu} \\
\zeta_{\mathcal{I}_\lambda}
\end{bmatrix}.
$$

As $a^\star_{i,m} = 0$ for any $(i, m) \in \mathcal{I}_\mu$ (and similarly, $a^\star_{i,m} = x_{i,m}$ for any $(i, m) \in \mathcal{I}_\nu$, $\sum_i a^\star_{i,m} = b_m$ for any $m \in \mathcal{I}_\lambda$), this can be further simplified as

$$
\underbrace{
\begin{bmatrix}
\boldsymbol{M}_1 & (\boldsymbol{M}_2)_{:,\mathcal{I}_\mu} & -(\boldsymbol{M}_2)_{:,\mathcal{I}_\nu} & -(\boldsymbol{M}_3)_{:,\mathcal{I}_\lambda} \\
\mathrm{diag}(\mu^\star_{\mathcal{I}_\mu}) & 0 & 0 & 0 \\
\mathrm{diag}(\nu^\star_{\mathcal{I}_\nu}) & 0 & 0 & 0 \\
\mathrm{diag}(\lambda^\star_{\mathcal{I}_\lambda})D_{\mathcal{I}_\lambda} & 0 & 0 & 0
\end{bmatrix}
}_{\boldsymbol{M}_{\mathcal{I}}}
\begin{bmatrix}
da \\
d\mu_{\mathcal{I}_\mu} \\
d\nu_{\mathcal{I}_\nu} \\
d\lambda_{\mathcal{I}_\lambda}
\end{bmatrix}
$$

$$
=
\begin{bmatrix}
\alpha - (\boldsymbol{M}_2)_{:,\mathcal{I}^c_\mu}d\mu_{\mathcal{I}^c_\mu} + (\boldsymbol{M}_2)_{:,\mathcal{I}^c_\nu}d\mu_{\mathcal{I}^c_\nu} + (\boldsymbol{M}_3)_{:,\mathcal{I}^c_\lambda}d\mu_{\mathcal{I}^c_\lambda} \\
\beta_{\mathcal{I}_\mu} \\
\gamma_{\mathcal{I}_\nu} \\
\zeta_{\mathcal{I}_\lambda}
\end{bmatrix}.
$$

Note that $|\mathcal{I}_\mu| + |\mathcal{I}_\nu| + |\mathcal{I}_\lambda| \geq NM - N$ when at least $NM - N$ constraints are tight. Since $x > 0$, the total number of linearly independent constraints on $da$ is therefore $N(\text{from } M_1) + (NM - M) = NM$, which uniquely determines $da$. Due to the diagonal structure of $\boldsymbol{M}_2$, the block diagonal structure of $\boldsymbol{M}_3$, and the fact that the optimal allocation $a^\star$ needs to satisfy $v_i^\top a_i^\star > 0$ for all $i$ to avoid making the objective function infinitely negatively large, it is not hard to see that we can determine the remaining variables $d\mu_{\mathcal{I}_\mu}$, $d\nu_{\mathcal{I}_\mu}$, and $d\lambda_{\mathcal{I}_\mu}$ from the first row block of $\boldsymbol{M}_{\mathcal{I}}$. The invertibility of $M$ leads to the uniqueness of $\boldsymbol{M}^{-\top}[(\frac{\partial \ell}{\partial a})^\top, \boldsymbol{0}^\top, \boldsymbol{0}^\top, \boldsymbol{0}^\top]^\top$, which obviously implies the differentiability of the mapping from $(v, x)$ to $a^\star$. When less than $NM - N$ constraints are tight, the number of linear constraints on $da$ is at most $NM - 1$, which cannot uniquely determine the $NM$-dimensional vector $da$.

$\square$

## C  Proof of Proposition 1

Under the assumptions of the proposition, $\boldsymbol{M}$ is an invertible matrix. The following equations follow from the fact that the utility function $u_i$ does not depend on the dual variables

$$
\frac{\partial \ell}{\partial v'_{1,1}} = (\frac{\partial \ell}{\partial a^\star})^\top \frac{\partial a^\star}{\partial v'_{1,1}} = \left[(\frac{\partial \ell}{\partial a^\star})^\top, \boldsymbol{0}^\top, \boldsymbol{0}^\top, \boldsymbol{0}^\top\right] \boldsymbol{M}^{-1} \boldsymbol{h}(e_1, \boldsymbol{0}, \boldsymbol{0})
$$

$$
= \boldsymbol{h}(e_1, \boldsymbol{0}, \boldsymbol{0})^\top \boldsymbol{M}^{-\top} \left[(\nabla_{a^\star} u_i)^\top, \boldsymbol{0}^\top, \boldsymbol{0}^\top, \boldsymbol{0}^\top\right]^\top.
$$

The important observation is that the computation of the gradient with respect to any entry in $v'_i$ (and also $x'_i$ and $w$) uses the product of $\boldsymbol{M}$ and $[\ (\nabla_{a^\star} u_i)^\top, \boldsymbol{0}^\top, \boldsymbol{0}^\top, \boldsymbol{0}^\top\ ]^\top$, and only the vector $\boldsymbol{h}$ is different. Taking advantage of this fact, it is not hard to see (following a line of analysis similar to derivation of Eq. (8) in Amos & Kolter (2017)) that once we solve

$$
\left[g_a^\top, g_\mu^\top, g_\nu^\top, g_\lambda^\top\right]^\top = \boldsymbol{M}^{-\top} \left[(\nabla_{a^\star} u_i)^\top, \boldsymbol{0}^\top, \boldsymbol{0}^\top, \boldsymbol{0}^\top\right]^\top.
$$

the gradients admit the closed-form expressions

$$\nabla_{v_{i'}} u_i = \big( - w_i I_{M \times M} - (\mu_i^\star - \nu_i^\star - \lambda^\star)(a_i^\star)^\top \big) g_{a,iM:(i+1)M},$$
$$\nabla_{x_{i'}} u_i = \text{diag}(\nu^\star) g_\nu, \quad \nabla_{w_i} u_i = v_i^\top g_{a,iM:(i+1)M}.$$

$\square$

## D  Derivation of Gradients of Exploitability-Averse Proportional Fairness Mechanism

With values $v$ and demands $x$, let $a^\star$ denote the solution of (9), i.e., $a^\star = f_\omega^{RPF}(v, x, b)$. The aim of this section is to derive the (sub)gradient of a loss function $\ell$ with respect to $z$, $v$, $x$, and $w$ when we are given $\nabla_{a^\star} \ell$. The arguments used here are mostly straightforward extension of the result in Sec. 4.

The equations in the KKT system of (9) include

$$-\frac{w_i v_i}{v_i^\top a_i^\star} + z_i - \mu_i^\star + \nu_i^\star + \lambda^\star = 0, \quad \forall i = 1, \cdots, N$$
$$\mu_{i,m}^\star a_{i,m}^\star = 0, \quad \forall i = 1, \cdots, N, \quad m = 1, \cdots, M$$
$$\nu_{i,m}^\star (a_{i,m}^\star - x_{i,m}) = 0, \quad \forall i = 1, \cdots, N, \quad m = 1, \cdots, M$$
$$\lambda_m^\star (Da^\star - b)_m = 0, \quad \forall m = 1, \cdots, M.$$

Since the optimal solution $a^\star$ need to satisfy $v_i^\top a_i^\star > 0$ for every agent $i$, the first equation can be simplified as

$$(\mu_i^\star - \nu_i^\star - \lambda^\star) v_i^\top a_i^\star + w_i v_i - z_i v_i^\top a_i^\star = 0, \quad \forall i = 1, \cdots, N.$$

Taking the differential,

$$(d\mu_i^\star - d\nu_i^\star - d\lambda^\star - dz_i) v_i^\top a_i^\star + (\mu_i^\star - \nu_i^\star - \lambda^\star - z_i) dv_i^\top a_i^\star$$
$$+ (\mu_i^\star - \nu_i^\star - \lambda^\star - z_i) v_i^\top da_i^\star + w_i dv_i + dw_i vi = 0, \forall i$$
$$d\mu_{i,m}^\star a_{i,m}^\star + \mu_{i,m}^\star da_{i,m}^\star = 0, \quad \forall i, m$$
$$d\nu_{i,m}^\star (a_{i,m}^\star - x_{i,m}) + \nu_{i,m}^\star (da_{i,m}^\star - dx_{i,m}) = 0, \quad \forall i, m$$
$$d\lambda_m^\star D_m a^\star + \lambda_m^\star D_m da^\star - d\lambda_m^\star b_m = 0.$$

This systems of equations can be written in the concise matrix form

$$M' \begin{bmatrix} da \\ d\mu \\ d\nu \\ d\lambda \end{bmatrix} = \begin{bmatrix} c' \\ \mathbf{0} \\ \text{diag}(\nu^\star) dx \\ \mathbf{0} \end{bmatrix},$$

where the matrix $M' \in \mathbb{R}^{(3NM+M) \times (3NM+M)}$ is

$$M' = \begin{bmatrix} M_1' & M_2 & -M_2 & -M_3 \\ \text{diag}(\mu^\star) & \text{diag}(a^\star) & 0 & 0 \\ \text{diag}(\nu^\star) & 0 & \text{diag}(a^\star - x) & 0 \\ \text{diag}(\lambda^\star) D & 0 & 0 & \text{diag}(Da^\star - b) \end{bmatrix},$$

the vector $c' \in \mathbb{R}^{NM}$ is

$$c' = c + \begin{bmatrix} v_1^\top a_1^\star dz_1 + z_1 (a_1^\star)^\top dv_1 \\ v_2^\top a_2^\star dz_2 + z_2 (a_2^\star)^\top dv_2 \\ \vdots \\ v_N^\top a_N^\star dz_N + z_N (a_N^\star)^\top dv_N \end{bmatrix},$$

the matrix $\boldsymbol{M}'_1 \in \mathbb{R}^{NM \times NM}$ is

$$\boldsymbol{M}'_1 = \begin{bmatrix} (\mu_1^\star - \nu_1^\star - \lambda^\star - z_1)v_1^\top & \cdots & 0 \\ \vdots & \ddots & \vdots \\ 0 & \cdots & (\mu_N^\star - \nu_N^\star - \lambda^\star - z_N)v_N^\top \end{bmatrix},$$

and the matrices $\boldsymbol{M}_2, \boldsymbol{M}_3$ and vector $\boldsymbol{c}$ are defined in Sec. 4.1.

Again, it can be shown that once we pre-compute and save

$$(\boldsymbol{M}')^\top \begin{bmatrix} g_a \\ g_\mu \\ g_\nu \\ g_\lambda \end{bmatrix} = \begin{bmatrix} \frac{\partial \ell}{\partial a^\star} \\ \mathbf{0} \\ \mathbf{0} \\ \mathbf{0} \end{bmatrix},$$

the gradients have the closed-form expressions

$$\frac{\partial \ell}{\partial v_i} = \left( -w_i I_{M \times M} - (\mu_i^\star - \nu_i^\star - \lambda^\star - z_i)(a_i^\star)^\top \right) g_{a,iM:(i+1)M},$$

$$\frac{\partial \ell}{\partial x} = \text{diag}(\nu^\star)g_\nu, \quad \frac{\partial \ell}{\partial w_i} = v_i^\top g_{a,iM:(i+1)M},$$

$$\frac{\partial \ell}{\partial z_i} = v_i^\top a_i^\star g_{a,iM:(i+1)M}.$$

## E  Proportional Fairness Mechanism and Discipline Parameterized Program

Discipline parameterized programs (DPP) are a special class of convex programs introduced in Agrawal et al. (2019), in which the authors show how (sub)gradients can be derived through the mapping from parameters of a DPP to its optimal solution. We start the discussion by introducing the DPP, which is an optimization program of the form

$$y^\star = \underset{y}{\text{argmin}} \quad f_0(x, \theta)$$

$$\text{s.t.} \quad f_i(y, \theta) \le 0, \quad i = 1, \ldots, m_{ineq}$$

where $y \in \mathbb{R}^{n_1}$ is the decision variable, $\theta \in \mathbb{R}^{n_2}$ is the parameter which we need to derive the gradient for, and the functions $f_i$ are convex. In addition, all functions are required to be **affine** in a special sense. An expression is said to be **parameter-affine** if it is affine in affine in the parameter $\theta$ and **variable-free** (does not involve variable $x$). An expression is said to be **parameter-free** if it does not involve any parameter (it can involve the variable $x$). Any expression $\phi_{\text{prod}}(z_1, z_2) = z_1 z_2$ as the product of $z_1 \in \mathbb{R}$ and $z_2 \in \mathbb{R}^p$ for any $p$ is affine if at least one of the two conditions is true:

- $y_1$ or $y_2$ is both parameter-free and variable-free

- among $y_1$ and $y_2$, one is parameter-affine and the other is parameter-free

When we try to capture (4) by a DPP, the decision variable $y$ corresponds to allocation $a$, and the parameter $\theta$ abstracts $(v, x, b, w)$. As part of the objective of (4), $a_i^\top v_i$ can be verified to be affine since $a_i$ is parameter-free and $v_i$ is parameter-affine. However, the overall objective $-\sum_{i=1}^N w_i \log(a_i^\top v_i)$ is not affine as $w_i$ is not parameter-free and $\log(a_i^\top v_i)$ is not variable-free, parameter-affine or parameter-free. While sometimes re-formulation can be made to transform a non-DPP to an equivalent DPP, such re-formulation is not possible in this case.

# F  Generalization Bound

## F.1  Definitions & Preliminaries

Let $\mathcal{F}$ denote a class of bounded functions $f : Z \to [-c, c]$ defined on an input space $Z$ for some $c > 0$. Let $D$ be a distribution over $Z$ and let $\mathcal{S} = \{z_1, z_2, \cdots, z_L\}$ be a sample drawn i.i.d from some distribution $D$ over input space $Z$.

**Definition 4.** *The capacity of the function class $\mathcal{F}$ measured in terms of empirical Rademacher complexity on sample $\mathcal{S}$ is defined as*

$$\hat{\mathcal{R}}(\mathcal{F}) := \frac{1}{L}\mathbb{E}_\sigma\left[\sup_{f \in \mathcal{F}}\sum_{z_i \in \mathcal{S}}\sigma_i f(z_i)\right],$$

*where $\sigma \in \{-1, 1\}^L$ and each $\sigma_i$ is drawn from a uniform distribution on $\{-1, 1\}$.*

**Definition 5.** *The covering number of a set $\mathcal{M}$, denoted as $\mathcal{N}_\infty(\mathcal{M}, \epsilon)$, is the minimal number of balls of radius $\epsilon$ (measured in the $l_{\infty,1}$ distance) needed to cover the set $\mathcal{M}$.*

For example, the $l_{\infty,1}$ distance between mechanisms $f, f' \in \mathcal{M}$ is given as

$$\max_{(v,x,b)}\sum_{i=1}^{N}\sum_{j=1}^{M}|f_{ij}(v, x, b) - f'_{ij}(v, x, b)|.$$

**Lemma F.1** ((Shalev-Shwartz & Ben-David, 2014))**.** *Then with probability at least $1 - \delta$ over draw of $S$ from $D$, for $f \in \mathcal{F}$,*

$$\mathbb{E}_{z \sim D}[f(z)] \leq \frac{1}{L}\sum_{l=1}^{L}f(z_l) + 2\hat{\mathcal{R}}_L(\mathcal{F}) + 4c\sqrt{\frac{2\log(4/\delta)}{L}}$$

**Lemma F.2** (Massart)**.** *Let $\mathcal{G}$ be some finite subset of $\mathbb{R}^m$ and $\sigma_1, \sigma_2, \cdots, \sigma_m$ be independent Rademacher random variables. Then,*

$$\mathbb{E}\left[\sup_{g \in \mathcal{G}}\frac{1}{m}\sum_{i=1}^{m}\sigma_i g_i\right] \leq \frac{\sqrt{2\left(\sup_g \sum_i g_i^2\right)\log|G|}}{m}.$$

Let $\phi : \mathbb{R}^N \mapsto \mathbb{R}^N$ represent the activation function of the any layer for input $s \in \mathbb{R}^{NM}$, given as

$$\phi = [\mathtt{softmax}(s_{1,1}, \cdots, s_{N,1}), \cdots, \mathtt{softmax}(s_{1,m}, \cdots, s_{N,M})],$$

where $\mathtt{softmax} : \mathbb{R}^N \mapsto [0, 1]^N$. Note that for any $u \in \mathbb{R}^N$,

$$\mathtt{softmax}_i(u) = \frac{e^{u_i}}{\sum_{k=1}^{N}e^{u_k}}.$$

**Lemma F.3** ((Dütting et al., 2023))**.** *For any $s, s' \in \mathbb{R}^{NM}$, the activation function for a $\mathtt{softmax}$ layer is $1-$Lipschitz, i.e.,*

$$\|\phi(s) - \phi(s')\|_1 \leq \|s - s'\|.$$

**Lemma F.4** ((Dütting et al., 2023))**.** *Let $\mathcal{F}_k$ be a class of feed-forward neural networks that maps an input vector $y \in \mathbb{R}^{d_0}$ to an output vector $\mathbb{R}^{d_k}$, with each layer $l$ containing $T_l$ nodes and computing $z \mapsto \phi_l(w^l z)$ and $\phi_l : \mathbb{R}^{T_l} \to [-\kappa, \kappa]^{T_l}$. Further, for each network in $\mathcal{F}_k$, let the parameter matrices $\|w^l\|_1 \leq W$ and $\|\phi_l(s) - \phi_l(s')\| \leq \Phi\|s - s'\|_1$ for any $s, s' \in \mathbb{R}^{T_{l-1}}$. The covering number of the network is*

$$\mathcal{N}_\infty(\mathcal{F}_k, \epsilon) \leq \left\lceil\frac{2\kappa d^2 W(2\Phi W)^k}{\epsilon}\right\rceil^d,$$

*where $T = \max_{l \in [k]} T_l$ and $d$ is the total number of parameters in the network.*

**Corollary F.4.1.** *For the same configuration as in Lemma F.4, but having $\|\phi_l(s) - \phi_l(s')\| \leq \|s - s'\|_1$ for all hidden layer activation functions ($l < k$), and having $\|\phi_o(s) - \phi_o(s')\| \leq \Phi\|s - s'\|_1$ for the output layer, the covering number is given as*

$$\mathcal{N}_\infty(\mathcal{F}_k, \epsilon) \leq \left\lceil\frac{\Phi d^2 \kappa(2W)^{k+1}}{\epsilon}\right\rceil^d.$$

### F.2 Proof of Theorem 2

Consider a parametric class of mechanisms, $f^\omega \in \mathcal{M}$, defined using parameters $\omega \in \mathbb{R}^d$ for $d > 0$. With a slight abuse of notation, let $u^\omega(v, x, b) := u(f^\omega(v, x, b), v, x)$ and let $y \mapsto (v, x, b)$, $y_i' \mapsto ((v_i', v_{-i}), (x_i', x_{-i}), b)$. Consider the following function classes

$$\mathcal{U}_i = \left\{ u_i^\omega : Y \mapsto \mathbb{R} \,\big|\, u_i^\omega(y) = \sum_m v_{i,m} \min\{f_{i,m}^\omega(y), x_{i,m}\} \text{ for some } f^\omega \in \mathcal{M} \right\}$$

$$\text{NSW} \circ \mathcal{M} = \left\{ f : Y \mapsto \mathbb{R} \,\big|\, f(y) = \sum_{i=1}^N \log u_i^\omega(y) \text{ for } u_i^\omega \in \mathcal{U}_i \right\}$$

$$\exp \circ \mathcal{U}_i = \left\{ e_i : Y \mapsto \mathbb{R} \,\big|\, e_i(y) = \max_{y_i'} u_i^\omega(y_i') - u_i^\omega(y) \text{ for } u_i^\omega \in \mathcal{U}_i \right\}$$

$$\overline{\exp} \circ \mathcal{U} = \left\{ h : Y \mapsto \mathbb{R} \,\big|\, h(y) = \sum_{i=1}^N e_i(y) \text{ for some } (e_1, e_2, \cdots, e_N) \in \exp \circ \mathcal{U} \right\}.$$

**Proposition 2.** *Suppose Assumption 1 holds. Let $\mathcal{M}$ denote the class of mechanisms and fix $\delta \in (0, 1)$. With probability at least $1 - \delta$, over draw of $L$ profiles from $F$, for any parameterized allocation $f^\omega \in \mathcal{M}$,*

$$\mathbb{E}_{y \sim F}\left[ \sum_{i=1}^N \log u_i^\omega(y) \right] \geq \frac{1}{L} \sum_{l=1}^L \sum_{i=1}^N \log u_i^\omega(y^{(l)}) - 2N\Delta_L - CN\sqrt{\frac{\log(1/\delta)}{L}},$$

*where $C$ is a distribution independent constant and*

$$\Delta_L = \inf_{\epsilon > 0} \left\{ \epsilon + N(\log \psi + 1)\sqrt{\frac{2\log(\mathcal{N}_\infty(\mathcal{M}, \frac{\epsilon}{\psi}))}{L}} \right\}.$$

*Proof.* Using Lemma F.1, the result follows except the characterization of the empirical Rademacher complexity that we derive below. By the definition of the covering number, we have that for any $h(y) \in \text{NSW} \circ \mathcal{M}$, there is a $\hat{h}(y) \in \widehat{\text{NSW}} \circ \mathcal{M}$ such that $\max_y |h(y) - \hat{h}(y)| \leq \epsilon$. We have the following

$$\hat{\mathcal{R}}_L(\text{NSW} \circ \mathcal{M}) = \frac{1}{L}\mathbb{E}_\sigma\left[ \sup_u \sum_{l=1}^L \sigma_l \sum_{i=1}^N \log u_i^\omega(y^{(l)}) \right]$$

$$= \frac{1}{L}\mathbb{E}_\sigma\left[ \sup_h \sum_{l=1}^L \sigma_l \hat{h}(y^{(l)}) \right] + \frac{1}{L}\mathbb{E}_\sigma\left[ \sup_u \sum_{l=1}^L \sigma_l \left\{ h(y^{(l)}) - \hat{h}(y^{(l)}) \right\} \right]$$

$$\leq \frac{1}{L}\mathbb{E}_\sigma\left[ \sup_{\hat{h}} \sum_{l=1}^L \sigma_l \hat{h}(y^{(l)}) \right] + \frac{1}{L}\mathbb{E}_\sigma \|\sigma\| \epsilon$$

The result follows from the following arguments:

i.) We shall first establish that

$$\mathcal{N}_\infty(\text{NSW} \circ \mathcal{M}) \leq \mathcal{N}_\infty(\mathcal{M}, \frac{\epsilon}{\psi}).$$

By definition of the covering number for the mechanism class $\mathcal{M}$, there exists a cover $\hat{\mathcal{M}}$ of size $|\hat{\mathcal{M}}| \leq \mathcal{N}_\infty(\mathcal{M}, \frac{\epsilon}{\psi})$ such that for any $f^\omega \in \mathcal{M}$ there is a $\hat{f}^\omega \in \hat{\mathcal{M}}$ such that for all $y$,

$$\sum_{i,m} |f_{i,m}^\omega(y) - \hat{f}_{i,m}^\omega(y)| \leq \frac{\epsilon}{\psi}$$

For $g(y) = \sum_{i=1}^N \log u_i^\omega(y)$, we have

$$\left| g(y) - \hat{g}(y) \right| = \left| \sum_{i=1}^N \left\{ \log u_i^\omega(y) - \log \hat{u}_i^\omega(y) \right\} \right|$$

$$\leq \psi \Big| \sum_{i=1}^{N} u_i^\omega(y) - \hat{u}_i^\omega(y) \Big|$$

$$\leq \psi \Big| \sum_i \sum_m v_{i,m} \Big\{ \min\{f_{i,m}^\omega(y), x_{i,m}\} - \min\{\hat{f}_{i,m}^\omega(y), x_{i,m}\} \Big\} \Big|$$

$$\leq \psi \sum_i \sum_m v_{i,m} \Big| \min\{f_{i,m}^\omega(y), x_{i,m}\} - \min\{\hat{f}_{i,m}^\omega(y), x_{i,m}\} \Big|$$

$$\leq \psi \sum_i \sum_m v_{i,m} \Big| f_{i,m}^\omega(y) - \hat{f}_{i,m}^\omega(y) \Big| < \epsilon.$$

ii.) We have from Massart's lemma (Lemma F.2),

$$\hat{\mathcal{R}}_L(\mathrm{NSW} \circ \mathcal{M}) \leq \sqrt{\sum_l \Big( \hat{h}(y^{(l)}) \Big)^2} \frac{\sqrt{2 \log(\mathcal{N}_\infty(\mathrm{NSW} \circ \mathcal{M}), \epsilon)}}{L} + \epsilon.$$

iii.) We have the trivial bound,

$$\sqrt{\sum_l \Big( \hat{h}(y^{(l)}) \Big)^2} \leq \sqrt{\sum_l \Big( \sum_i \log u_i^\omega(v^{(l)}) + n\epsilon \Big)^2} \leq N(\log \psi + 1)\sqrt{L}.$$

The result follows. $\qquad\square$

**Proposition 3.** *Suppose Assumption 1 holds. Let $exp_i(\omega) := \mathbb{E}_y\Big[ \max_{y'} u_i^\omega(y_i') - u_i^\omega(y) \Big]$ and $\widehat{exp}_i(\omega) := \frac{1}{L} \sum_{l=1}^{L} \Big[ \max_{\bar{y}} u_i^\omega(\bar{y}_i^{(l)}) - u_i^\omega(y^{(l)}) \Big]$. Under the same assumptions as in Theorem 2, the empirical exploitability satisfies the following:*

$$\frac{1}{N} \sum_{i=1}^{N} exp_i(\omega) \leq \frac{1}{N} \sum_{i=1}^{N} \widehat{exp}_i(\omega) + \Delta_L^e + C'\sqrt{\log(1/\delta)/L},$$

*where $C'$ is a distribution independent constant and*

$$\Delta_L^e = \inf_{\epsilon > 0} \left\{ \epsilon + (2\psi + 1)N\sqrt{\frac{2\log(\mathcal{N}_\infty(\mathcal{M}, \frac{\epsilon}{2N}))}{L}} \right\}.$$

*Proof.* As before, using Lemma F.1, the result follows except the characterization of the empirical Rademacher complexity of the class $\hat{\mathcal{R}}(\overline{\exp} \circ \mathcal{U})$, which we do below. The proof builds on the following results.

i.) $\mathcal{N}_\infty(\exp \circ \mathcal{U}_i, \epsilon) \leq \mathcal{N}_\infty(\mathcal{U}_i, \frac{\epsilon}{2})$.

By the definition of covering number $\mathcal{N}_\infty(\mathcal{U}_i, \epsilon)$, there exists a cover $\hat{\mathcal{U}}_i$ with size at most $\mathcal{N}_\infty(\mathcal{U}_i, \frac{\epsilon}{2})$ such that for any $u_i^\omega \in \mathcal{U}_i$ there is a $\hat{u}_i^\omega \in \hat{\mathcal{U}}_i$ with

$$\max_y |u_i^\omega(y) - \hat{u}_i^\omega(y)| \leq \frac{\epsilon}{2}.$$

We have for the exploitability for each agent $i$ with any $y$,

$$|\max_{y_i'} u_i^\omega(y_i') - u_i^\omega(y) - \max_{y_i'} \hat{u}_i^\omega(y_i') + \hat{u}_i^\omega(y)|$$

$$\leq |\max_{y_i'} u_i^\omega(y_i') - \max_{y_i'} \hat{u}_i^\omega(y_i')| + |u_i^\omega(y) - \hat{u}_i^\omega(y)| \leq \epsilon.$$

where the last inequality follows from the following relation.

$$\max_{\bar{y}_i} u_i^\omega(\bar{y}_i) = u_i^\omega(y^*)$$

$$\leq \hat{u}_i^\omega(y^*) + \epsilon/2 \leq \hat{u}_i^\omega(\bar{y}^*) + \epsilon/2 \leq \max_{\bar{y}} \hat{u}_i^\omega(\bar{y}) + \epsilon,$$

$$\max_{y_i} \hat{u}_i^\omega(y_i) = \hat{u}_i^\omega(y^*)$$

$$\leq u_i^\omega(y^*) + \epsilon/2 \leq u_i^\omega(\bar{y}^*) + \epsilon/2 = \max_{\bar{y}} u_i^\omega(\bar{y}) + \frac{\epsilon}{2}.$$

ii.) $\mathcal{N}_\infty(\overline{\exp} \circ \mathcal{U}, \epsilon) \leq \mathcal{N}_\infty(\mathcal{U}, \frac{\epsilon}{2N})$. The result following by considering a cover $\mathcal{N}_\infty(\mathcal{U}, \frac{\epsilon}{2N})$ such that

$$\max_y \sum_{i=1}^N |u_i^\omega(y) - \hat{u}_i^\omega(y)| \leq \frac{\epsilon}{2N},$$

and using the same arguments as above.

iii.) $\mathcal{N}_\infty(\mathcal{U}, \epsilon) \leq \mathcal{N}_\infty(\mathcal{M}, \epsilon)$.

By definition of the covering number for the mechanism class $\mathcal{M}$, there exists a cover $\hat{\mathcal{M}}$ of size $\hat{\mathcal{M}} \leq \mathcal{N}_\infty(\mathcal{M}, \epsilon)$ such that for any $f^\omega \in \mathcal{M}$ there is a $\hat{f}^\omega \in \hat{\mathcal{M}}$ such that for all $y$,

$$\sum_{i,m} |f_{i,m}^\omega(y) - \hat{f}_{i,m}^\omega(y)| \leq \epsilon.$$

We have for the class $\mathcal{U}$,

$$\left| \sum_{i=1}^N u_i^\omega(y) - \hat{u}_i^\omega(y) \right| \leq \left| \sum_i \sum_m v_{i,m} \left\{ \min\{f_{i,m}^\omega(y), x_{i,m}\} - \min\{\hat{f}_{i,m}^\omega(y), x_{i,m}\} \right\} \right|$$

$$\leq \sum_i \sum_m v_{i,m} \left| \min\{f_{i,m}^\omega(y), x_{i,m}\} - \min\{\hat{f}_{i,m}^\omega(y), x_{i,m}\} \right|$$

$$\leq \sum_i \sum_m v_{i,m} \left| f_{i,m}^\omega(y) - \hat{f}_{i,m}^\omega(y) \right| < \epsilon.$$

The result follows from Lemma F.2 and similar arguments as in Proposition 2. $\qquad \square$

Note that the activation functions on all hidden layers are ReLU functions, which are 1-Lipschitz. The output layer of the regularized PF activation is Lipschitz from Assumption 1 with some constant $\Phi > 0$.

Using the definitions of $\Delta$'s with the activation having a Lipschitz constant of $\Phi$, and the hidden layers having a Lipschitz constant of 1, and with $d = \max\{K, MN\}$ in Corollary F.4.1, we have from Proposition 2 and Proposition 3 that the following holds with probability at least $1 - \delta$

$$\max\{\varepsilon_{\mathbf{logNSW}}(f^\omega, L), \varepsilon_{\mathbf{exp},i}(f^\omega, L)\} \leq \mathcal{O}\left( \psi N \frac{\sqrt{Rd \log(LN\Omega\Phi \max\{K, MN\})}}{L} + N\sqrt{\frac{\log(1/\delta)}{L}} \right).$$

# G   Proof of Theorem 3

It is obvious that the exploitability of any mechanism at agent $i$ cannot exceed the utility of agent $i$ when its demands are fully satisfied. This means for any mechanism $f$ and $v \in \mathcal{V}, x \in \mathcal{D}, b \in \mathcal{B}$, we have

$$\mathbf{expl}_i(f, v, x, b) \leq \sum_{m=1}^M v_{i,m} x_{i,m} \leq M \overline{v}\,\overline{x}, \tag{20}$$

where the second inequality is due to the boundedness of values and demands.

Let $p_F$ and $p_{F'}$ denote the probability density function associated with $F$ and $F'$. We have for any $i$

$$E_{(v,x,b)\sim F}[\mathbf{expl}_i(f, v, x, b)]$$

$$= E_{(v,x,b)\sim F'}[\mathbf{expl}_i(f,v,x,b)] + \Big( E_{(v,x,b)\sim F}[\mathbf{expl}_i(f,v,x,b)] - E_{(v,x,b)\sim F'}[\mathbf{expl}_i(f,v,x,b)] \Big)$$

$$\leq \epsilon + \int \mathbf{expl}_i(f,v,x,b)\,(p_{F'}(v,x,b) - p_F(v,x,b))\,dv\,dx\,db$$

$$\leq \epsilon + \int |\mathbf{expl}_i(f,v,x,b)|\,|p_F(v,x,b) - p_{F'}(v,x,b)|\,dv\,dx\,db$$

$$\leq \epsilon + M\,\overline{v}\,\overline{x}\int |p_F(v,x,b) - p_{F'}(v,x,b)|\,dv\,dx\,db$$

$$= \epsilon + 2M\,\overline{v}\,\overline{x}d_{TV}(F,F'),$$

where the third inequality applies (20) and the final equation comes from the definition of TV distance, i.e., for any distribution $F_1, F_2$ over $\mathcal{X}$

$$d_{TV}(F_1, F_2) = \frac{1}{2}\int_{\mathcal{X}} |p_F(x) - p_{F'}(x)|\,dx.$$

$\square$

