# OpenReview forum: "Regularized Proportional Fairness Mechanism for Resource Allocation Without Money"
_TMLR — Accepted by TMLR_

### Review · Reviewer_qw7W · 2024-10-11

**Summary Of Contributions:**

The authors present a neural network architecture that implements an approximately truthful mechanism for maximizing proportional fairness (aka Nash Social Welfare) when allocating resources, without the use of money.  It is known that truthful bidding in this setting cannot be attained without inefficient allocations (e.g., by "money burning").  This paper aims to come as close to efficiency as possible while satisfying an approximate truthfulness criterion: that no bidder can improve their allocation very much by lying about their valuation (i.e., their demand and value for each good).

The proposed system is based on regularization: if we knew the bidders' true valuations in the first place, we could reduce their incentive to lie by nudging each allocation in the opposite direction of their incentive for misreporting.  (i.e., if they bid in a certain way because to increase the allocation of good 1, then artificially reduce their allocation of good 1 in order to make the lie not be worth it).  The auctioneer doesn't know the valuations, so instead a neural network is trained to output an estimated optimal-misreport direction, which will then be used to regularize an optimization that minimizes the regularized NSW.

**Audience:**

Yes

**Claims And Evidence:**

Yes

**Requested Changes:**

My main concerns with this paper revolve around the reporting of some of the empirical results.  I think that these issues will be straightforward to address but will substantially improve the paper.

* Table 1: Are the +/- ranges standard deviations, or 95% CIs, or what?
* Figure 5: I don't really see support here for the claim that RPF-Net "determines allocations in a smoother way" than PF; can you elaborate on what features of this figure lead you to say that?
* Why does Table 2 give training and inference time for a *specific* ten-agent three-resource problem?  It would be far better to report on the performance on a distribution of such problems and give averages and error bounds (similar to Table 1).

=== Minor issues ===

Here are some additional nitpicks and minor issues.

* p.3: Zeng et al. appear to be /very/ similar; you allude to differences in the "activation function" throughout the paper, but it would be very helpful to more explicitly describe the differences early on (most relevantly that they don't need to solve an optimization problem in their output layer).
* p.3: It's not really accurate to say that [Shoham & Leyton-Brown, 2008] "design sophisticated monetary transfer schemes"; this is a textbook.
* p.3: 0.841 > 3/4, so how do Guo and Conitzer "nearly" match 0.841 if Cole et al. prove a bound of 3/4 for $n=2$?
* p.5: "By contrast, RPF-Net proposed in this work significantly reduces the exploitability.": Are you claiming statistical significance?  If so, based on what test?  If not, it is better to use a synonym such as "materially" or "substantially" to avoid confusion.
* p.9: Possible typo: $vh(dv,dx,dw)$
* p.13: "which shares the same training objective as RPF-Net": This is somewhat misleading, since ExS-Net doesn't include regularization in its objective, not so?
* p.14: "the heavy-tailed Cauchy distribution, which is often used to model unobserved strategic behaviors (Taleb, 2010)": Is this claim really supported in "Black Swans"?  Can you give the relevant passage?
* Figure 5: Why are all 4 subfigures labelled "RPF-Net"?

**Strengths And Weaknesses:**

One difficulty of this approach is that the optimal misreport depends on the result of the optimization; thus, the gradient of the loss needs to include the outcome of the optimization.  A key technical contribution of the paper is deriving the relationship between the gradient of the optimization's KKT conditions and those of the loss function, allowing for an overall gradient of the loss in terms of the parameters of the neural network.

Overall, this is an exciting and innovative application of the differentiable economics approach to mechanism design.  I think that the gradient translation approach has the potential to be very influential in future work on mechanism design. The technique is developed and described in a very clear way.  The empirical results are largely compelling.  I recommend that the paper be accepted with minor revisions.

---

> ### Author Response · Authors · 2024-11-06
> **Response to Reviewer qw7W**
>
> We are grateful of the efforts that the reviewer dedicated to evaluating the paper and the recognition of our contributions. We have carefully incorporated your comments and questions in the revision. Please find the response to your comments below.
>
> 1) Regarding what numbers are reported in Table 1, we confirm that these are standard deviations. We now clearly indicate what the numbers are in the caption of Table 1.
>
> 2) >Figure 5: I don't really see support here for the claim that RPF-Net "determines allocations in a smoother way" than PF; can you elaborate on what features of this figure lead you to say that?
>
> We thank the reviewer for catching the issue with the figure interpretability. There was a typo in the titles of the top figures, which is now corrected. The titles of the top 2 figures should be "PF Mechanism -- Allocation of Resource 1 to Agent 1" and "PF Mechanism -- Allocation of Resource 2 to Agent 1" (which match Figure 5 caption). Across any column, the bottom plot (generated by RPF-Net) resembles but has a smoother decision boundary than the top plot (generated by PF mechanism).
>
> 3) >Why does Table 2 give training and inference time for a specific ten-agent three-resource problem? It would be far better to report on the performance on a distribution of such problems and give averages and error bounds (similar to Table 1).
>
> We thank the reviewer for catching this, which is due to a mis-wording. These numbers are obtained indeed on a distribution of problems (same setting as the one that produces Figure 3, data generated according to Eq.(17)). We have now corrected the table caption to say that we obtain the number from a specific problem dimension and a specific distribution of training and testing samples (rather than a specific problem).
>
> 4) We thank the reviewer for suggesting more discussion of Zeng et al. (2024) early on in the paper. We have made the following changes to the last few sentences of the second paragraph of Section 1.1.
>
> '''
> Also assuming the access to training samples, Zeng et al. (2024) is highly related to our work. Zeng et al. (2024) studies learning an approximately fair and IC mechanism named ExS-Net parameterized by a neural network and trains the network parameters on the same objective that we consider in this work. Our important distinction from ExS-Net lies in the activation function of the output layer: while the activation function in ExS-Net is composed of a simple softmax function and a synthetic agent which receives the portion of resources to be withheld, the activation function we propose leverages a convex optimization program. ExS-Net is an important baseline for comparison. In Section 6 we show that the proposed RPF-Net mechanism materially outperforms ExS-Net due to the innovation in network architecture.
> '''
>
> 5) We thank the review for pointing out the inaccurate description of Shoham & Leyton-Brown (2008). We have replaced it with a more proper reference Chawla et al. (2010).
>
> 6) >0.841 > 3/4, so how do Guo and Conitzer "nearly" match 0.841 if Cole et al. prove a bound of 3/4 for $n=2$
>
> Our statement should be made more precise. The bounds in Guo and Conitzer (2010) (including worst-case lower bound and almost matching algorithm performance) are established for the two-agent and two-resource setting, whereas the bound established by Cole et al. (2013) is to for $n$ agents with the number of items being a manipulable parameter chosen to construct the worst-case scenario. When $n=2$, Cole et al. (2013) shows that a competitive factor of more than 0.75 cannot be achieved in the worst case when the number of resources **becomes large**. Since the 0.841 bound in Guo and Conitzer (2010) is for 2 resource, there is no contradiction. We have refined our description of the two papers and greatly appreciate the reviewer's help in correcting our earlier inaccurate statement.
>
> 7) >By contrast, RPF-Net proposed in this work significantly reduces the exploitability... it is better to use a synonym such as "materially" or "substantially" to avoid confusion.
>
> We thank the reviewer for pointing out the inaccuracy in our statement. We do not claim statistical significance and have revised the wording.
>
> 8) Regarding $vh$, we thank the reviewer for catching the typo and have corrected it.
>
> 9) >"which shares the same training objective as RPF-Net": This is somewhat misleading, since ExS-Net doesn't include regularization in its objective, not so?
>
> Yes, ExS-Net does not employ any regularization term in the **network architecture** while RPF-Net does, but ultimately both ExS-Net and RPF-Net are trained to maximize NSW subject to a constraint on exploitability, i.e. the training objective is Eq. (10).

---

> ### Author Response · Authors · 2024-11-06
> **Response to Reviewer qw7W Continued**
>
> 10) >"the heavy-tailed Cauchy distribution, which is often used to model unobserved strategic behaviors (Taleb, 2010)": Is this claim really supported in "Black Swans"? Can you give the relevant passage?
>
> We thank the reviewer for catching the issue in the reference. We wanted to cite a different paper of Taleb. We have revised the reference and sentence itself (attached below).
>
> '''
> We would like to draw the perturbation from a heavy-tailed distribution to increase the likelihood of generating extreme values (Taleb, 2020). The Cauchy distribution has been used to explain extreme events like Flash Crash (Parker, 2016) and to describe price fluctuations (Casault et al., 2011), which motivates our selection.
> '''
>
> We hope our response addresses your concerns. Please let us know if you have any further comments and/or questions that we can answer.
>
> References
>
> Casault, S., Groen, A.J. and Linton, J.D., 2011. On the use of the Cauchy distribution to describe price fluctuations in R&D and other forms of real assets. In 8th ESU Conference on Entrepreneurship (2011), p 1-9. Universidad de Sevilla.
>
> Chawla, S., Hartline, J.D., Malec, D.L. and Sivan, B., 2010, June. Multi-parameter mechanism design and sequential posted pricing. In Proceedings of the forty-second ACM symposium on Theory of computing (pp. 311-320).
>
> Parker, E., 2016. Flash Crashes: The role of information processing based subordination and the Cauchy distribution in market instability. Journal of Insurance and Financial Management, 2(7).
>
> Taleb, N.N., 2020. Statistical consequences of fat tails: Real world preasymptotics, epistemology, and applications. arXiv preprint arXiv:2001.10488.

---

### Review · Reviewer_W3sP · 2024-10-16

**Summary Of Contributions:**

This work studies the resource allocation problem for divisible items without payment, aiming to improve the Nash Social Welfare (NSW) while reducing exploitability. The authors propose a novel method, RPF-Net, based on a neural network that approximates the misreport value and incorporates a linear regularization designed to enhance robustness against potential misreports. From a theoretical perspective, the authors demonstrate how to calculate the gradient for this complex network and provide a square-root decreasing rate for the generalization error with a finite dataset. Additionally, experimental simulation results support the efficiency of the proposed method.

**Audience:**

Yes

**Claims And Evidence:**

Yes

**Requested Changes:**

See Weaknesses.

**Strengths And Weaknesses:**

Strengths:

1. The authors present a novel method, RPF-Net, which effectively balances Nash Social Welfare (NSW) and potential exploitability.

2. In terms of computational complexity, the authors demonstrate how to compute the gradient, and the algorithm shows lower complexity compared to baseline methods.

3. Experimental simulation results confirm the efficiency of the proposed method.

1. In Section 3, the assumption that the misreport follows a greedy policy with respect to the mechanism in (6) is questionable, especially when agents only have partial information. Under such circumstances, predicting the misreport value through a neural network becomes challenging.

2. From a theoretical perspective, the authors only provide a convergence rate for the generalization error with a finite dataset. However, it is well-known that as the dataset size increases, the generalization error tends to decrease to zero. A more critical issue may be the approximation error from the neural network, which might not approach zero even with an infinite dataset. It is important to discuss whether the neural network can accurately approximate the misreport value and whether gradient descent can find such a network.

3. In the simulation experiments, it is unclear how the misreport value for adversarial agents in (19) is determined, as it appears the mechanism $u_i$ also depends on the misreport value. Additionally, in Table 1, the partial allocation mechanism only achieves about 1/4 of the NSW compared to other methods, which contradicts the related work suggesting that each agent in the PA mechanism should receive at least 1/e of the utility in the corresponding PF mechanism.

---

> ### Author Response · Authors · 2024-11-06
> **Response to Reviewer W3sP**
>
> We thank the reviewer for the feedback and for the time and efforts spent on evaluating our work. Please find below some clarifying remarks on your main comments.
>
> 1) >In Section 3, the assumption that the misreport follows a greedy policy with respect to the mechanism in (6) is questionable, especially when agents only have partial information. Under such circumstances, predicting the misreport value through a neural network becomes challenging.
>
> We thank the reviewer for bringing the question up and want to clarify a possible misunderstanding here. We do not make any assumptions on the misreport. Equation (6) merely serves as a motivation for developing the form of regularization introduced in Eq.(8). It is true that the worst-case misreport can only be computed by an agent when it has the knowledge of the mechanism and the valuations of other agents. When the designed mechanism accounts for this worst-case setting, we know that the degree to which the mechanism can be exploited is only lower if the agents have less information.
>
> To support the argument above, we note that the performance of the mechanism is investigated when the training data is generated by agents reporting their parameters i) truthfully, ii) randomly, iii) adversarially to PF mechanism, and it was observed that the performance of the learned mechanism  consistently achieves a good trade-off between NSW and exploitability across all cases.
>
> 2) >From a theoretical perspective ... It is important to discuss whether the neural network can accurately approximate the misreport value and whether gradient descent can find such a network.
>
> We agree with the reviewer that approximation error is indeed an important consideration for mechanisms learned from data. We note that, as in the related literature on differentiable economics, one cannot guarantee that gradient descent on the RHS of Eq.(3) with respect to $v_i’,x_i$ finds the maximizer, especially when the mechanism is a complicated mapping encoded by a neural network. Also, while we aim to optimize the objective (10), we cannot guarantee the parameter $\omega$ obtained after training is a feasible solution. If the desired exploitability threshold $\epsilon$ is made too small, we observe that the solution may sometimes not exactly satisfy the constraint. These are important issues that need further investigation and are out of scope of the current paper, whose main aim is to extend the applicability of the PF mechanism by providing a means to evaluate and improve its exploitability.
>
> 3) >In the simulation experiments, it is unclear how the misreport value for adversarial agents in (19) is determined, as it appears the mechanism also depends on the misreport value.
>
> We thank the reviewer for helping us identify a typo in Eq.(19), which is now corrected. It should read $v_{i}^l,x_{i}^l=\arg\max_{v_i',x_i'}u_i(f^{PF}((v_i',\bar{v}\_{-i}^l),(x_i',\bar{x}\_{-i}^l),b),\bar{v}^l,\bar{x}^l)$, i.e. the samples are generated when the agents believe that a PF mechanism is used to make allocation and act adversarially accordingly, as we explain in the paragraph above Eq.(19).
>
> 4) >in Table 1, the partial allocation mechanism only achieves about 1/4 of the NSW compared to other methods, which contradicts the related work suggesting that each agent in the PA mechanism should receive at least 1/e of the utility in the corresponding PF mechanism.
>
> Note that PA mechanism guarantees that **each agent** receives at least $1/e$ of the PF allocation, whereas PF mechanism is defined as the **product** of all agents' utilities. With two agents, the worst case NSW of PA mechanism is guaranteed to be only $1/e^2$ of the NSW of PF mechanism, which is about 0.135 and lower than 1/4.
>
> We hope our response addresses your concerns. Please let us know if you have any further comments and/or questions that we can answer.

---

### Review · Reviewer_abuk · 2024-11-03

**Summary Of Contributions:**

The paper is a contribution to a growing line of research on “differentiable economics”: the use of rich function approximators and optimization tools adapted from deep learning to solve problems in mechanism design.

The particular focus is on resource allocation *without money*. So, the goal is to allocate some goods to agents who value those goods differently. As in all mechanism design, the problem is that the agents might misreport their values to manipulate the outcome in their favor, so the mechanism must be made strategyproof. The performance goal is Nash social welfare (product of agent utilities), which formalizes a notion of fairness.

In this setting, the proportional fairness (PF) mechanism directly computes the Nash social welfare optimizing allocation, but is not strategyproof. (A redundant but important point: this means that a strategyproof mechanism can’t be optimal for this problem.) There is a well-known strategyproof mechanism (Partial Allocation), but it involves deliberately withholding goods and is therefore suboptimal.

What the authors want to do is explore mechanisms that are somehow “in between” these — to (informally) make a Pareto improvement by trading off a little Nash Social Welfare to get a lot less IC violation.

They do this using a neural network as a function approximator, using a roughly similar recipe to Dütting et al., though with significant technical innovations for their problem.

At a high level, this recipe is that a neural network will map agent reports to an allocation, and will be trained to maximize performance. Its last-layer activation should enforce problem constraints as much as possible. Incentive compatibility should be enforced approximately by having a penalty term in the loss function, and IC violations can be measured by optimizing bidder reports also via a gradient-based method.

For this problem, the “activation” on the neural network is a regularized version of the proportional fairness problem. So another interpretation of what they do is run a proportional fairness mechanism, with a learned, bidder-dependent regularization term. Both training the neural network and estimating exploitability then require optimizing with respect to the *solution* of that optimization.

It’s relatively well-known that one can “backprop through an optimization problem” in order to enforce complex or unusual constraints on neural network outputs. Doing it for this particular problem requires the authors to come up with significant technical innovations, though.

Putting all this together, the authors have their RPF-Net architecture and training algorithm, which they run on a bunch of plausible mechanism design problems and show works better than theoretical baselines. A key point is also that it’s possible to visualize and make sense of RPF-Net’s outputs, so it is a tool for exploring the space of mechanisms and not just for optimizing performance.

There are also some nice extra results, including showing robustness to distribution shift under the TV metric, generalization results following those in Dütting et al.

**Audience:**

Yes

**Broader Impact Concerns:**

There are no broader impact concerns.

**Claims And Evidence:**

Yes

**Requested Changes:**

The only real reservation I have with the paper is not so important — it’s the last paragraph. I think this paragraph reads oddly given that “differentiating through optimization programs” is something ML researchers have been interested in for many years at this point. The idea is not original and people are well aware of it. (However, this paper’s technical contributions to this long line of research are original.) I think the paragraph could just be safely deleted.

Two papers that should definitely be cited:

"Learning revenue-maximizing auctions with differentiable matching", which does auction design by differentiating through an LP. The technical details are very different (Sinkhorn not inverting KKT system) but similar concept

https://proceedings.mlr.press/v151/curry22a.html

"Deep learning for two-sided matching": another example of using differentiable economics to explore tradeoffs in a mechanism design situation where there is an impossibility result, though on a completely different problem (Gale-Shapley style matching). For them, it is trading off between stability and IC, as opposed to Nash social welfare and IC.

https://arxiv.org/abs/2107.03427

Very weak (not required) request: it would be nice to emphasize very clearly somewhere that exploitability measurements are only empirical. I don't think this counts against a paper like this, but it is sometimes a point of confusion with theorists/economists.

**Strengths And Weaknesses:**

Strengths:

- Their work on differentiating through their optimization problem is nice and is in fact probably of interest even to people who don’t care at all about mechanism design.
- The distributional robustness result is both very cool and useful. I don’t think I have seen this type of result in a differentiable economics paper before.
- The experiments are quite solid and convincing. No complaints here.
- Ability to visualize the mechanism is important, as the main motivation of this work is exploring the space of mechanisms of interest to theorists (vs. actually learning some mechanism that notionally would be fielded).

Weaknesses:

- There aren’t really substantial weaknesses to the paper. It’s a very solid contribution to this line of work.
- The regret/exploitability measurements are only empirical. This is very common and not a deal-breaker, but it would be even better if there were some true guarantees on IC.

---

> ### Author Response · Authors · 2024-11-06
> **Response to Reviewer abuk**
>
> We are grateful of the efforts that the reviewer dedicated to evaluating the paper and the recognition of our contributions. Your feedback crucially shapes our revision and the directions we plan to pursue following up on this work.
>
> 1)
> >The regret/exploitability measurements are only empirical... it would be even better if there were some true guarantees on IC.
>
> We agree with the reviewer’s comment that the desiderata of the learned mechanisms are only verified empirically. To our knowledge, true guarantees are so far established only for hand-designed mechanisms. It would certainly be a valuable future direction to explore the theoretical guarantees provided by neural mechanisms, even on simple architectures and problem settings. We have added the following remark to the end of Section 3 noting the empirical nature of the work to avoid any confusion.
>
> '''
> Remark 1. We note that our work measures and optimizes exploitability only empirically and cannot guarantee strict exploitability constraint satisfaction, at least for two reasons. First, we cannot guarantee that gradient descent applied to the right hand side of Eq.(3) with respect to $v_i’,x_i'$ finds the global maximizer, especially when the mechanism is a complicated mapping encoded by a neural network. Second, while we aim to optimize the objective (10), we cannot guarantee that the parameter $\omega$ obtained from training is a feasible solution. If the desired exploitability threshold $\epsilon$ is chosen too small, we sometimes observe constraint violation.
> '''
>
> 2) >The only real reservation I have with the paper is not so important — it’s the last paragraph... I think the paragraph could just be safely deleted.
>
> We thank the reviewer for the suggestion and have removed the paragraph.
>
> 3) We thank the reviewer for suggesting the highly relevant references, which we somehow missed in the initial draft. We have added a discussion of these works in Section 1.1.
>
> We hope our response addresses your concerns. Please let us know if you have any further comments and/or questions that we can answer.

---

### Decision · Action_Editor_csvw · 2024-12-05

**Recommendation:** Accept as is

**Comment:**

The reviewers all agree that the paper clearly meets the TMLR criteria and would be of pretty broad interest: the approach of differentiating through an optimization problem would be of general interest beyond even just their results in mechanism design.  All of this makes an easy decision.

**Audience:**

Yes, definitely.

**Claims And Evidence:**

Yes.